

# Amplified potential for vegetation stress under climate change-induced intensifying compound extreme events in the Greater Mediterranean Region

Patrick Olschewski[1], Mame Diarra Bousso Dieng[1], Hassane Moutahir[1,2], Brian Böker[1], Edwin Haas[1], Harald Kunstmann[1,3], and Patrick Laux[1,3]

[1]Institute of Meteorology and Climate Research (IMK-IFU), Karlsruhe Institute of Technology, Campus Alpin, Kreuzeckbahnstraße 19, 82467 Garmisch-Partenkirchen, Germany
[2]Department of Ecology, University of Alicante, 03690 Sant Vicent del Raspeig, Alicante, Spain
[3]Institute of Geography, University of Augsburg, Alter Postweg 118, 86159 Augsburg, Germany

**Correspondence:** Patrick Olschewski (patrick.olschewski@kit.edu)

**Abstract.** The Mediterranean Basin is one of the regions most affected by climate change. It is highly dependent on the impact of climate change on agricultural efficiency and food security. While rising temperatures and decreasing precipitation levels already impose great risks, the effects of compounding extreme events (CEEs) can be significantly more severe and amplify the risk. It is therefore of high importance to assess these risks under climate change on a regional level to implement efficient

adaption strategies. This study focuses on False Spring Events (FSEs), which impose a high risk of crop losses during the beginning of the vegetation growing period, as well as Heat and Drought-based CEEs (HDCEs) in summer, for a high-impact future scenario (RCP8.5). The results for 2070-2099 are compared to 1970-1999. In addition, deviations of the near-surface atmospheric state under FSEs and HDCEs are investigated with the aim to improve the predictability of these events. We apply a multivariate, trend-conserving bias correction method (MBCn) accounting for temporal coherency between the inspected

variables derived from EUR-CORDEX. This method proves to be a suitable choice for the assessment of percentile threshold-based CEEs. The results show a potential increase in frequency of FSEs for large portions of the study domain, especially impacting later stages of the warming period, caused by disproportionate changes in the behavior of warm phases and frost events. Frost events causing FSEs predominantly occur under high-pressure conditions and northerly to easterly wind flow. HDCEs are projected to significantly increase in frequency, intensity, and duration, mostly driven by dry, continental air masses.

This intensification is multiple times higher than that of the univariate components. This study improves the understanding of the unfolding of climate change in the Mediterranean and shows the need for further, locally refined, investigations and adaptation strategies.

## 1 Introduction

The latest Assessment Report by the Intergovernmental Panel on Climate Change (IPCC) on the current climate status has

yet again made strikingly clear, that the consequences of human actions significantly impacted weather and climate on a global scale and continue to do so (Eyring et al., 2021). Among the most significant impacts of ongoing climate change is





the intensification of extreme events, especially in terms of temperature and precipitation (Seneviratne et al., 2021). These changes are projected to include a significant increase in the frequency and intensity of heat-related extremes, as well as intensified heavy precipitation events, but also an amplified risk of agricultural drought. Additional consequences of human-

induced global warming also include negative impacts on agriculture. As stated by Bezner Kerr et al. (2022), there is high confidence that climate change is imposing stress on, among others, agriculture and forestry. Furthermore, the authors point out extreme events to significantly impair food security and heat and drought to be among the main drivers exacerbating the risk of production losses.

   While the general understanding of these changes is high on a global level, the level of confidence in these projections

varies on a regional to local level. Giorgi (2006) conducted an investigation of regions with the highest exposure to the effects of climate change (Climate Change "Hot-Spots") and demonstrated the Mediterranean, which includes southern Europe as well as parts of northern Africa and western Asia, to be among the two most vulnerable regions worldwide. As this region is characterized by a high population density and high socioeconomic importance, the effects of climate change must be monitored closely to enable suitable protective and mitigative measures.

In order to assess the potential effects of climate change, general circulation models (GCM) have been extensively used in the past. These models, driven on a global basis, are capable of providing large-scale information on changes in atmospheric predictors, although their spatial resolution is generally low. While GCMs are able of providing reasonable results under specific circumstances, the quality of GCM output proves to be too low to be applied on a sub-global scale (Di Virgilio et al., 2022; Yang and Villarini, 2021; Wang et al., 2021; Hardiman et al., 2008). To improve the quality of these projections,

regional climate models (RCM) driven by the GCM output can be consulted, by conducting simulations on a regional to local level and in a higher temporal and spatial resolution (Giorgi, 2019). While this procedure, also known as dynamical downscaling, can improve the quality of climate projections, systematic model bias may still be inherited, originating from both, the GCM and RCM (Eden et al., 2012). To overcome this limitation, the inherited bias from climate models can be statistically corrected according to a reference data set of a higher quality. This method is referred to as bias correction and can,

for example, be performed using station observations, satellite data, or reanalysis. As (Cannon, 2018; Vrac and Friederichs, 2015; Gudmundsson et al., 2012) have proven, performing statistical bias correction can significantly reduce deviations of one or multiple predictors from a reference data set and therefore improve the reliability of climate projections.

   Compound extreme events (CEE) have moved more and more into the scientific focus, as the joint effects of multiple hazards may surpass those of univariate extremes (Zscheischler et al., 2020). In the context of extreme events potentially exacerbating

risks to agriculture, water availability, and food security (Bezner Kerr et al., 2022), high demand for robust projections of CEE in regions with a high vulnerability regarding these aspects becomes evident. Therefore, this study aims at investigating two types of compound events with detrimental effects on agriculture and vegetation in the Greater Mediterranean Region (GMR), a global climate change "Hot-Spot". The types of CEE analyzed in this study include False Spring Events (FSEs), which are defined as a freezing event occurring after the start of the crop-related growing season (SGS) (Ault et al., 2013; Gu et al., 2008).

Freezing during this highly vulnerable period in the early stages of plant development may cause significant damage, resulting in yield loss or failure. Recently, FSEs have been investigated mostly in moderate, mid-latitudinal climate zones. For example,





disproportionate changes in the last day of frost (LDF) during spring and the SGS have been shown for Central Europe (Vitasse et al., 2018; Zohner et al., 2016) and the United States (Peterson and Abatzoglou, 2014). Chen et al. (2021) demonstrated a negative correlation between the SGS date and mean air temperature over temperate China, indicating an earlier SGS under

warmer conditions. Therefore, if the risk of spring freezing events does not proportionally decline, the risk of FSEs increases (Labe et al., 2017; Inouye, 2008; Gu et al., 2008). Less focus has been put on subtropical climate zones, where the risk of frost events is considerably lower than in moderate climate zones. However, as the GMR is characterized by complex topography, with high mountain ridges adjoining wide lowlands that represent a transition zone between the GMR and cool-temperate Central Europe, the effects of freezing conditions may potentially reach out to the subtropical parts of the domain. A variety of

methods to estimate the LDF and SGS exist. One example is to collect and assess crop-specific empirical data, where specific thresholds for freezing and growing conditions can be taken into account (Chamberlain et al., 2019). This is often applied in regionalized studies or for specific plant species. In terms of heat-and-drought compound events (HDCEs), many studies investigated historical periods and demonstrated an increasing trend regarding the frequency and intensity of HDCEs (Ionita et al., 2021; Vogel et al., 2021). Fewer studies have investigated whether these changes will be persistent under future climate.

For example, Ruffault et al. (2020) demonstrated heat-and-drought-related weather conditions favoring the ignition of wildfires to likely be increased under climate change.

In this study, we aim at presenting a large-scale overview of the potential of FSEs occurring in the GMR. Therefore, we apply a simplified method derived from Peterson and Abatzoglou (2014) and Leeper et al. (2021) that uses generalized thermal thresholds to determine the LDF and SGS. In terms of HDCEs, we adopt an approach by Ionita et al. (2021), using the 3-

monthly standardized precipitation index (SPI-3) for drought indexing and percentile-based thresholds for daily maximum temperature. By conducting this study, we seek to increase the knowledge of how the effects of global warming will unfold in terms of FSEs and HDCEs in the GMR, regarding both the frequency and duration of these events.

Next to the projected changes in FSEs and HDCEs, we also investigated the deviations of crucial near-surface atmospheric predictors from the mean state during these events, in order to improve the level of knowledge on what conditions act favorable

towards the occurrence of FSEs and HDCEs. Similar works have been done by Ionita et al. (2021) and Mastrantonas et al. (2021), who investigated connections between extreme/compound events and large-scale atmospheric patterns. The inclusion of weather patterns, however, is aggravated in the context of multivariate bias correction, as the conservation of spatial and temporal coherence of multiple variables requires a significantly higher computational effort (Cannon, 2018). By considering sea level pressure as well as zonal ($u$) and meridional ($v$) wind speeds in an only temporally coherent bias correction setup,

we seek to reduce the computational cost while preserving the ability to assess the origin of air masses and estimate pressure deviations under FSEs and HDCEs.

In order to obtain these results, we apply a multivariate bias correction method based on the N-dimensional probability density function transform (MBCn) presented by Cannon (2018). 13 GCM-RCM combinations were obtained from CORDEX, as well as ERA5 data as reference for the bias correction procedure. We inspect a late-century future period (2070-2099) under

the high-impact RCP scenario 8.5 and compare the results to 1970-1999. The choice of scenario was made to demonstrate





potential changes in FSEs and HDCEs on the extreme end of the range of emission scenarios (Riahi et al., 2011). These research questions are addressed in the following sections:

a) To what extent can the quality of climate model output be improved, in the context of reproducing threshold-based metrics for a multivariate and interdependent task? b) Are FSEs relevant in the GMR, and how is their occurrence projected to change

towards the end of the $21^{st}$ century? What deviations in the near-surface atmospheric state are connected with FSEs?

c) How will the frequency, intensity, and duration of HDCEs change towards the end of the $21^{st}$ century? What deviations in the near-surface atmospheric state are connected with HDCEs?

d) What implications will the projected changes in FSEs and HDCEs potentially have on vegetation and crop efficiency?

## 2   Study domain

In this study, we investigate the potential effects of climate change on the Greater Mediterranean Region (GMR). The domain margins are given as 10° W - 37° E and 27.75° N - 50° N and are displayed in Figure 1. The GMR includes most parts of the southern European continent, as well as portions of northern Africa and western Asia. As a result of the Alpine orogeny, this region is characterized by a variety of mountainous formations that merge directly into the Mediterranean Sea. Some of the most prominent ridges are the Alps, the Apennines, and the Pyrenees on the European side, the Atlas formation on the African

side, and the Taurus on the Western Asian side. However, these mountain ranges are separated by wide lowlands, where large river systems drain precipitated water from the mountains toward the sea. These lowlands, e.g. the valleys of Ebro, Rhône, and Po, substantially differ from the mountain ranges in terms of climatic characteristics, adding even more to the diversity and complexity of the GMR.

This climatic complexity is also reflected in the Köppen-Geiger Climate Classification (Kottek et al., 2006). The northern

parts of the study domain are under the influence of the interplay between the polar front and subtropical highs within the zone of Westerlies (Cf climate, see Fig. 1). This causes a constant alteration between moderate temperatures and a high likelihood for precipitation under the maritime influence and dry air masses of continental origin, causing hot temperatures in summer and the opposite in winter. With decreasing latitude, the influence of the Westerlies recedes and the subtropical ridge becomes dominant. This persistent high-pressure system causes calm, warm, and dry conditions under descending air masses. Due

to the southward shift of the subtropical ridge in accordance with the annual cycle of the Intertropical Convergence Zone (ITCZ), the influence of the Westerlies becomes stronger in winter, causing precipitation levels to increase. The resulting warm and summer-dry climate (Cs climate), which is most common on the west sides of continents, is also often referred to as Mediterranean Climate. The southernmost area of the study domain, where the subtropical ridge is persistently dominant throughout the year and precipitation is rare, is classified as dry-arid, or dry-semi-arid (BW and BS climate). With increasing

elevation in the northern mountainous regions, the climate is cooler and more continental, with increased temperature spans and persistently high precipitation (Df climate). The highest portions of the Alps, where the mean temperature of the warmest months is below 10 °C, fall under the Polar-Tundra class (ET, Beck et al. (2018)).

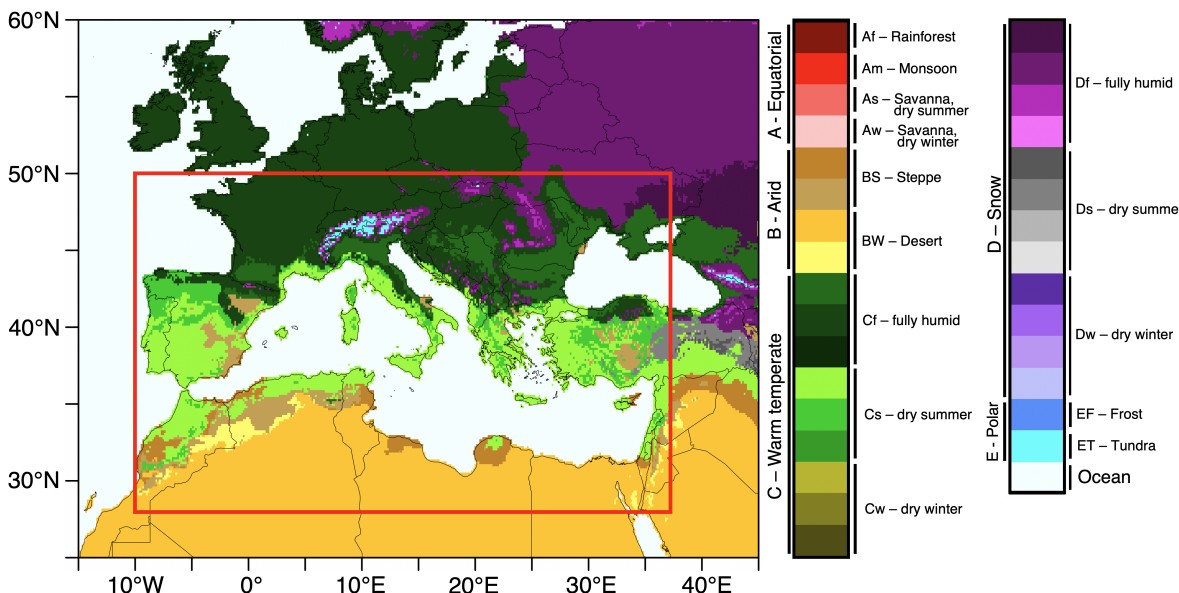

**Figure 1.** Study domain covering the Greater Mediterranean Region (GMR) outlined in red and classification of the local climate according to an updated version of the Köppen-Geiger Climate Classification. Data and visualization obtained from Kottek et al. (2006).

## 3 Data and Methods

To perform bias correction, three types of climate data are necessary. Initially, climate model output data for a historical period

must be obtained, next to the corresponding projection data for a future period under a specific scenario. In order to optimize the quality of the model output, a reference data set inheriting a higher accuracy regarding the "true" atmospheric state must be obtained. Depending on the research aim this can be, for example, observational station data, satellite data, or reanalysis. The latter is especially relevant in regions with a low density of observations, for example, regions with no or very low population density or regions over the ocean. While a variety of observational data sets is available for the land portions of our study

domain, it was our aim to specifically uphold the possibility of analyzing data over the Mediterranean Sea in future studies. Therefore, the latest reanalysis data set published by the European Centre for Medium-Range Weather Forecasts (ECMWF), ERA5 (Hersbach et al., 2020), was obtained for this study. In terms of model output, 13 different combinations of GCMs and RCMs were obtained from the **CO**ordinated **R**egional Climate **D**ownscaling **EX**periment (CORDEX), each consisting of dynamically downscaled realizations driven by GCMs obtained from the fifth phase of the Coupled Model Intercomparison

Project, CMIP5, initiated by the World Climate Research Programme (WCRP, Taylor et al. (2012)). This data was downloaded from the DKRZ node of the ESGF data portal (Cinquini et al., 2014).



## 3.1 Data

ERA5 data is used as reference data in this study and obtained from the Copernicus Climate Data Store (Hersbach et al., 2023).
Two versions of ERA5 reanalysis exist, globally covered standard ERA5 using a 0.25 x 0.25° resolution, and ERA5-Land,
which has a higher resolution of 0.1 x 0.1° (Muñoz Sabater, 2019). Standard ERA5 was chosen, as ERA5-Land does not cover
areas over the ocean. Out of the wide variety of variables included in ERA5, hourly 2 m air temperature, precipitation sum,
sea level pressure, as well as 10 m u- and v-components of wind speed (eastward and northward, respectively) were obtained
for this study and aggregated to daily mean values, respectively daily sums for precipitation. Hourly 2 m temperature was
aggregated to maximum and minimum daily values. ERA5 data was obtained for the period 1970-2020, however, the 30-year
period 1970-1999 is used as historical reference data within the bias correction process and denoted as HIST in the following.

The 13 model runs were each obtained from the European realization of the CORDEX ensemble, EUR-CORDEX (Cin-
quini et al., 2014), in the highest available spatial resolution of 0.11 x 0.11 km and on a daily temporal resolution. While
multiple RCP scenarios are provided, we focus only on the high-impact scenario RCP 8.5 (Riahi et al., 2011), and the distant
future period 2070-2099. Before being handed to the bias correction procedure, the model data was bilinearly interpolated
to match the grid structure of the ERA5 data. The 13 model combinations consist of 6 GCMs, including CNRM-CERFACS-
CNRM-CM5 (Voldoire et al., 2013), ICHEC-EC-EARTH (Hazeleger et al., 2012), IPSL-CM5A-MR (Dufresne et al., 2013),
MOHC-HadGEM2-ES (Collins et al., 2011), MPI-ESM-LR (Jungclaus et al., 2013; Stevens et al., 2013), and NCC-NorESM1-
M (Bentsen et al., 2013). As well as 4 RCMs, including DMI-HIRHAM5 (Christensen et al., 2007), GERICS-REMO2015
(Jacob et al., 2012; Jacob, 2001), KNMI-RACMO22E (van Meijgaard et al., 2008), and SMHI-RCA4 (Strandberg et al., 2014;
Samuelsson et al., 2011). All data utilized for this study is described in detail in Table 1.

## 3.2 Methods

In order to obtain optimized climate model output, we apply a multivariate and dependency-preserving bias correction method
(MBCn). This is necessary to uphold the temporal statistical relationship between the inspected variables, ensuring an im-
proved reflection of the "true" atmospheric relationship. The method of bias correction, and in specific the MBCn method,
is presented below. Regarding the definition of FSEs and HDCEs, we adopt threshold-based approaches that have been ap-
plied in former studies (Peterson and Abatzoglou, 2014; Leeper et al., 2021; Ionita et al., 2021). For both types of compound
events, a fixed prerequisite condition is defined, i.e. the last day of frost for FSEs and agricultural drought for HDCEs, and
the compounding element, i.e. the start of the growing season for FSEs and persistent heat for HDCEs, is additionally applied
based on multiple thermal thresholds of increasing intensity. In addition to changes in the frequency of FSEs and HDCEs,
we aim at offering insights into the prevailing near-surface atmospheric conditions during these events. Therefore, in addition
to minimum/maximum temperatures and precipitation, we included sea level pressure into the bias correction procedure, as
well as u- and v-components of near-atmospheric wind, to calculate the mean wind direction. The additional information on
pressure deviations can give hints on the prevailing type of action center (i.e. high pressure or low-pressure systems) and the
characteristics of the influencing air masses can be derived by means of their origin, i.e. the predominant wind direction. To





**Table 1.** Description of all obtained data sets for this study, including 13 GCM-RCM combinations from EUR-CORDEX for historical and future periods, as well as ERA5 reanalysis data for reference purposes.

| Climate model output (EUR-CORDEX) | | | historical | future |
| --- | --- | --- | --- | --- |
| GCM | RCM | abbreviation | period | period |
| CNRM-CERFACS-CNRM-CM5 | KNMI-RACMO22E | CNRM-CM5_RACMO22E | 1950 - 2005 | 2006 - 2100 |
| | SMHI-RCA4 | CNRM-CM5_RCA4 | 1970 - 2005 | 2006 - 2100 |
| ICHEC-EC-EARTH | DMI-HIRHAM5 | EC-EARTH_HIRHAM5 | 1951 - 2005 | 2006 - 2100 |
| | KNMI-RACMO22E | EC-EARTH_RACMO22E | 1950 - 2005 | 2006 - 2100 |
| | SMHI-RCA4 | EC-EARTH_RCA4 | 1970 - 2005 | 2006 - 2100 |
| IPSL-CM5A-MR | SMHI-RCA4 | IPSL_RCA4 | 1970 - 2005 | 2006 - 2100 |
| MOHC-HadGEM2-ES | DMI-HIRHAM5 | HadGEM2_HIRHAM5 | 1951 - 2005 | 2006 - 2099 |
| | KNMI-RACMO22E | HadGEM2_RACMO22E | 1950 - 2005 | 2006 - 2099 |
| | SMHI-RCA4 | HadGEM2_RCA4 | 1970 - 2005 | 2006 - 2099 |
| MPI-ESM-LR | SMHI-RCA4 | MPI-ESM-LR_RCA4 | 1970 - 2005 | 2006 - 2100 |
| NCC-NorESM1-M | DMI-HIRHAM5 | NorESM1_HIRHAM5 | 1951 - 2005 | 2006 - 2100 |
| | GERICS-REMO2015 | NorESM1_REMO2015 | 1950 - 2005 | 2006 - 2100 |
| | SMHI-RCA4 | NorESM1_RCA4 | 1970 - 2005 | 2006 - 2100 |
| **Reference data (reanalysis)** | | | | |
| ERA5 | - | - | 1940 - present | - |

inspect the statistical significance of near-atmospheric deviations under CEEs from the mean state, we apply a Mann-Whitney U test (Mann and Whitney, 1947; Student, 1908). As the 13 included models are independent, this test is applied separately to each of the models. The sum of models indicating statistical significance is obtained for both, positive and negative deviations from the mean, and presented in the results. As the inspection of mean wind directions is aggravated, due to a break in the scale between 359° and 0°, we inspect the mode instead of the mean, i.e. the predominant wind direction with the highest frequency within the sample. The statistical significance of the deviations of wind direction is obtained by applying a Fisher Exact test (Fisher, 1935) to the occurrence count distributions of the 8 wind directions N, NE, E, SE, S, SW, W, and NW. For each test, we consider statistical significance at a significance level of 5%.

### 3.2.1 Univariate bias correction

When aiming at an optimization of climate model output, multiple approaches of differing complexity exist. For example, if a linear bias towards a reference data set is to be removed, a delta (additive), or a factor (multiplicative), can be added to the model output (Deque, 2007). In general, the bias-corrected time series of a variable $x_{bc}$ can be obtained, when a statistical



transformation function $f$ is applied to the raw model output $x_{m,p}$, expressed as

$$x_{bc} = f(x_{m,p}), \tag{1}$$

by Piani et al. (2010). However, complex climate models often inherit a more complex bias structure, when e.g. trends and
instationarities come into play. To specifically account for differing bias within the distribution of a simulated climate variable,
the method of empirical quantile mapping (EQM) was introduced (Piani et al., 2010; Boé et al., 2007; Gudmundsson et al.,
2012). Within EQM, the cumulative distribution function (CDF) $F$ of the raw model output $m$ within a historical calibration
period $c$ is matched to the corresponding CDF of the reference data (e.g. reanalysis) $o$ within $c$, given as

$$F_{m,c}(x_{m,c}) = F_{o,c}(x_{o,c}) \tag{2}$$

by Gudmundsson et al. (2012) and Tong et al. (2021). In the next step, the CDF of the raw model output within the projected
period $p$ is matched to the inverse CDF of the reference data in $c$, $F_{o,c}^{-1}$, as given in

$$x_{bc} = F_{o,c}^{-1}[F_{m,p}(x_{m,p})], \tag{3}$$

in order to receive the bias corrected model data $x_{bc}$ (Gudmundsson et al., 2012; Tong et al., 2021). In a more general sense,
the additional "knowledge" of the reference data regarding the distribution structure of the "true" atmospheric state is applied
to the raw model output for a known, historical period. By doing so, information on the performance of the correction can be
evaluated. If the correction performance is deemed as satisfactory, and under the assumption that model bias is stationary within
the historical and projection periods (Maraun, 2012; Maraun et al., 2010), the same additional "knowledge" of the reference
data is applied to the projection period.

Trends within the raw model output may have a negative effect on correction performance and, respectively, trends infused
by the correction method may not represent the "true" atmospheric changes, as pointed out by Maurer and Pierce (2014) and
Maraun (2013). To account for this, the method of EQM was adjusted to initially extract trends from the raw model output, then
perform the correction process, and afterward apply the trend back to the corrected model output, as introduced by Cannon
et al. (2015). Within this Quantile Delta Mapping (QDM) procedure, the linear trend $\Delta_m$ regarding each time step $t$, defined
as

$$\Delta_m(t) = \frac{x_{m,p}(t)}{F_{m,c}^{-1}[F_{m,p}^t[x_{m,p}(t)]]}, \tag{4}$$

is returned to the corrected model output by applying

$$\hat{x}_{bc,\Delta}(t) = \hat{x}_{bc}(t)\Delta_m(t). \tag{5}$$

### 3.2.2 Multivariate bias correction

All previously described methods are applied independently to each variable. While these methods are not capable of specif-
ically adjusting the day-to-day variability to match that of the reference data, the temporal physical coherence is nevertheless





upheld, allowing for long-term climatological inspections (Olschewski et al., 2023). However, the independence of the correction process for multiple variables, where each variable is corrected on its own, aggravates the investigation of multivariate matters such as compound events (Zscheischler et al., 2019; Rocheta et al., 2014). Therefore, methods accounting for the dependency of multiple variables become necessary. As for univariate bias correction, a variety of approaches towards mul-

tivariate bias correction has been developed, each differing, for example, in the statistical metric that is used to adjust the dependence, or the level of restriction due to specific assumptions (Vrac and Thao, 2020; Cannon, 2016; Vrac and Friederichs, 2015; Bürger et al., 2011). This study applies a multivariate bias correction procedure (MBCn) developed by Cannon (2018), based on an image processing technique using the N-dimensional probability density function transform (N-pdft) presented by Pitie et al. (2005); Pitié et al. (2007). Within MBCn, a random orthogonal rotation is first applied to the input climate data and

QDM is applied to the marginal distributions of the rotated input data. Finally, the QDM-adjusted data is inversely rotated to receive the multivariately corrected output (Cannon, 2018). As the author describes subsequently, this process is iterated until the distributions of the model data and the reference data match. In this study, we carried out 100 iterations, which has proven sufficient in previous studies (Dieng et al., 2022; Cannon, 2018). The suitability and potential of this method in terms of climate data and its application have been proven in multiple studies, including Cannon (2018), Dieng et al. (2022), Lemus-Canovas

and Lopez-Bustins (2021), Singh et al. (2021), and Meng et al. (2022).

### 3.3 Compound event definitions

A comprehensive overview of types and definitions of compound weather and climate events is provided by Zscheischler et al. (2020). In general, the authors discriminate four different types of compound events, based on their temporal and spatial characteristics. FSEs, as defined for this study, are representative of preconditioned compound events, in which the precondition

is the warm anomaly of daily minimum temperature, which is beneficial towards the onset of the growing season, and the hazard is a subsequent frost event, potentially causing damage to crops within the early stages of plant development. Due to its subsequent nature, FSEs may also be categorized as temporally compounding events. FSEs consist of anomalies of one variable only, i.e. daily minimum temperature, and are therefore considered as univariate compound events. Heat-and-Drought compound events (HDCEs), by definition in this study, consist of multiple hazards occurring simultaneously, which is classified

by Zscheischler et al. (2020) as a multivariate compound event. However, if persistent drought is considered a precondition, HDCEs may also be treated as a preconditioned compound event.

Percentile-based thresholds are a crucial component in this study. These thresholds were calculated for the historical period and subsequently applied to the future period. By applying historical thresholds to future data, the results show changes in extremes that have already been experienced and can therefore be better assessed by potential users. All included variables, as

well as extreme event and compound event definitions, are summarized in Table 2 and described in detail below.

### 3.3.1 False Spring Events

In order to define FSEs, we take three thresholds into account, including the start of the growing season (SGS), the number of days between SGS and the frost event, as well as the thermal definition of frost events. In terms of the SGS, Robeson (2002)





**Table 2.** Variables, extreme events, and compound event definitions utilized in this study.

| Name | Abbr. | Description | Source |
|---|---|---|---|
| **Variables** | | | |
| Day of year | DOY | | |
| Daily maximum temperature | $T_x$ | | |
| Daily minimum temperature | $T_n$ | | |
| Daily precipitation sum | PR | | |
| Daily mean sea level pressure | SLP | | |
| Daily mean wind direction | DIR | | |
| **Metrics** | | | |
| Last day of frost | LDF | Last DOY in spring period with $T_n$ below -2.2 °C | Peterson and Abatzoglou (2014) |
| Start of growing season | SGS | First DOY in spring period with $T_n$ above 0 °C, 5 °C, 10 °C for seven consecutive days | Leeper et al. (2021) |
| Standardized Precipitation Index | SPI-3 | 3-monthly Standardized Precipitation Index | McKee et al. (1993) |
| Heatwave | HW | Six or more consecutive days with $T_x$ above 90., 95., 99. percentile | Perkins and Alexander (2013) Ionita et al. (2021) |
| **Compound events** | | | |
| False Spring Event | FSE | DOY of SGS minus DOY of LDF < 0 | Peterson and Abatzoglou (2014) |
| False Spring Event Index | FSEI | Number of years within HIST with an FSE | Peterson and Abatzoglou (2014) |
| | $FSEI_0$ | FSEI with SGS threshold of 0 °C | |
| | $FSEI_5$ | FSEI with SGS threshold of 5 °C | |
| | $FSEI_{10}$ | FSEI with SGS threshold of 10 °C | |
| Heat-Drought Compound Event | HDCE | Day under HW conditions and SPI-3 < -1 | Ionita et al. (2021) |
| | $HDCE_{90}$ | HDCE with HW threshold at the 90. percentile | |
| | $HDCE_{95}$ | HDCE with HW threshold at the 95. percentile | |
| | $HDCE_{99}$ | HDCE with HW threshold at the 99. percentile | |

suggested the period within spring and fall freezes, i.e. the SGS to correspond to the day of the year (DOY), from which daily
minimum temperatures are persistently above 0 °C. Leeper et al. (2021) built upon this definition and suggested the use of
multiple thermal thresholds, i.e. 0 °C, 5 °C, and 10 °C, and these thresholds are also applied in this study. These thresholds can
be considered representative of different phases within the period of continuous warming.

Applying a time delay between the SGS and the FSE-defining frost event is an attempt to consider the most vulnerable
time of leaf tissue, in the phase between budburst and full leafout (Chamberlain et al., 2019). In this context, Peterson and
Abatzoglou (2014) applied various numbers of days between 0 and 15, of which the authors found no sensitivity of the results





to lag times above 7 days. In accordance with the authors, we adapted the time lag of seven days, as well as the definition of daily minimum temperatures below -2.2 °C for frost events.

Based on Peterson and Abatzoglou (2014), we investigate changes in the False Spring Event index (FSEI), which indicates the portion of years with an occurrence of FSE within a selected period. A year is counted towards FSEI, when the SGS, which

is defined as a period of at least seven days in which the daily minimum temperature does not fall below 0 °C, 5 °C, or 10 °C, happens prior to the last frost event, defined as a daily minimum temperature below -2.2 °C. The considered variants of FSEI are therefore dependent on the definition of the SGS and include $FSEI_0$, $FSEI_5$, and $FSEI_{10}$. In general, the data from January through June was considered in the calculation of FSE. However, the occurrence of FSEs is dependent on the DOY of the SGS, which lies within this range of months for the predominant part of the region (see section 4.2.1).

### 3.3.2 Heat-and-Drought compound events

Ionita et al. (2021) conducted an investigation of historical compound hot and dry events over Europe and their approach is adopted in this study. Initially, drought conditions are defined using the Standardized Precipitation Index (SPI, McKee et al. (1993)). The SPI takes the standardized accumulated monthly rainfall amount into account and fits it to a Gamma distribution, resulting in a mean of 0 and a standard deviation of 1. Values below zero, therefore, describe conditions with below-average

accumulated rainfall, whereas values above zero depict wetter-than-average conditions. As we focus on agricultural drought, we choose the 3-monthly aggregated version, i.e. SPI-3. According to Edwards and McKee (1997), drought conditions are present when the SPI-3 is lower than -1, respectively when the precipitation level of the preceding 3 months is at least one standard deviation lower than average. This is also applied to this study.

Derived from Perkins and Alexander (2013) and Ionita et al. (2021), we apply a percentile-based threshold to define heatwave

events (HW). We define an HW to be present when the daily maximum temperature exceeds the monthly $90^{th}$, $95^{th}$, $99^{th}$ percentiles for at least six consecutive days. I.e., the sixth day of a period for which this condition is true counts as one towards the HW statistic. If the heat period has a duration of nine consecutive days, days number six, seven, eight, and nine will count towards HW, as the precondition is given for all of these days. Correspondingly, a Heat-Drought Compound Event (HDCE) is defined as the joint occurrence of HW and an SPI-3 below -1, and every calendar day, for which the HW and SPI-3 conditions

are given, counts towards the HDCE statistic. As the hazard potential of HDCEs is highest during the summer months, we analyzed HDCEs for the months of June, July, and August.

## 4 Results

Firstly, the performance of the MBCn method regarding the overall long-term climatology and specifically the estimation of percentile-based thresholds is evaluated. This is a crucial component to estimate the quality of the projections of FSEs and

HDCEs. Subsequently, the prerequisites, the historical and projected frequencies of FSEs and HDCEs, and the corresponding near-surface atmospheric deviations linked to these events are presented.



## 4.1 Bias correction performance

In terms of long-term climatology for the spatial mean, it becomes clear from Fig. 2 that the regionally downscaled output from CORDEX still inherits a substantial amount of bias regarding ERA5. For the annual mean of daily maximum temperature, (Fig. 2a), the raw output of the 13 models lies within a range of roughly 2.5 °C, with most of the models underestimating the value indicated by ERA5. After applying MBCn, the 30-year mean value of all models, and therefore also the model mean, align with the ERA5 mean at 19.5 °C. As the daily minimum temperature inherits a close statistical relationship to the daily maximum temperature, the performance of bias correction is similar, and the results were moved to Appendix A. For the annual precipitation sum (Fig. 2b), the uncertainty range of the raw output spans almost 200 mm, with an equal number of models over- and underestimating the sum given in ERA5. However, the bias of overestimating models is higher than the bias of underestimating models, leading to a slight positive bias in the model mean of around 75 mm. After performing MBCn, the bias is significantly reduced for all models. While some models show a slight underestimation of the annual rainfall amount after MBCn, the uncertainty range of all models is reduced to under 10 mm.

As absolute and percentile-based thresholds are crucial components of the CEE definitions used in this study, we also inspected the ability of MBCn to align the distribution of percentiles derived from the daily data to that of the reference data. As can be seen in Fig. 2c for daily maximum temperature, the quality of raw CORDEX varies between models. In extreme cases, the mean absolute error is more than 1.5 °C. After MBCn percentile distributions of the models are aligned perfectly, which may also be expected from a quantile-fitting method. The indicated percentile values for the MBCn-corrected projection data (illustrated by colored circles) demonstrate the significant changes in daily maximum temperature that are projected by the models, with increases of up to 7 °C. For daily precipitation, the error margin of the raw model output spans up to 0.4 mm. After MBCn, the bias is significantly reduced and lies within a range between 0 and 0.01 mm. The projection data for the future period indicates no clear direction of change, but the majority of models shows a reduction in daily precipitation for moderate precipitation events and an increase in precipitation intensity for extreme precipitation events.

## 4.2 False Spring Events

### 4.2.1 Last day of frost (LDF) and start of growing season (SGS)

Crucial indicators for changes in the possibility of FSEs are changes in the LDF and the SGS, which are displayed in Fig. 3. In general, three gradients can be derived. Firstly, LDF and SGS tend to happen earlier in the year with decreasing latitude, which is consistent with an increasing temperature level under a decreasing solar zenith angle. In addition, LDF and SGS tend to happen earlier in the year in maritime regions under a moderate, oceanic influence, whereas more continental areas show a later occurrence. Also, LDF and SGS tend towards later occurrences with increasing altitude, due to decreasing temperature levels. Throughout the low-level areas of southern Europe, the LDF mostly occurs in March or early April. Moving southward, the LDF occurs earlier, around early February, and reaches late January in southern Portugal/Spain, northern Africa, and western Asia. In mountainous areas, depending on altitude, the LDF tends to occur between May and early June. All areas have a projected decrease in the DOY of the LDF within the future period in common. This decrease is mostly around 20 to 30





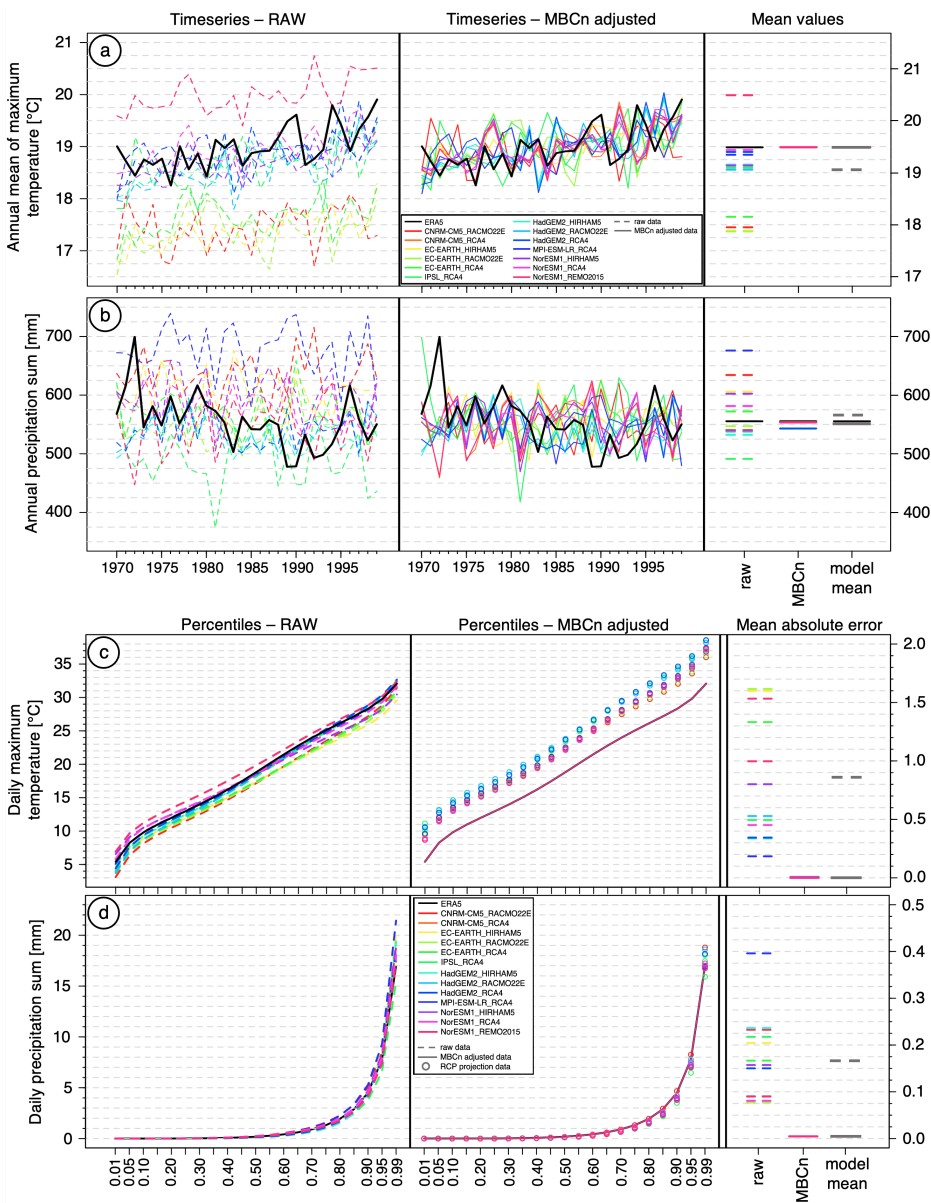

**Figure 2.** Evaluation of MBCn performance for long-term climatology and daily-based percentiles. a) Long-term time series of annual mean maximum temperature for ERA5 and 13 CORDEX models in raw (left), MBCn-corrected output (middle), and 30-year mean values (right). b) same as a), but for annual precipitation sum. c) percentile-based distributions of daily maximum temperature for ERA5 and 13 CORDEX models in historical raw data (left), MBCn-corrected historical and projected output (middle), and mean absolute error (right). d) same as c), but for daily precipitation.



days, or around one month, for most parts of southern Europe, corresponding to increasing temperature levels (Fig. 2). With increasing altitude, the DOY decrease reaches its spatial maximum of around 1.5 months. In coastal southern European areas, northern Africa, and western Asia, where frost events are in general rare, the reduction is less prominent.

The SGS with a temperature threshold of 0 °C ($SGS_0$, Fig. 3b and f) occurs particularly early in the year in most western and southern parts of the domain, except for mountainous regions. Moving east, the date increases from mid-to-late January to early March. In elevated areas, the $SGS_0$ starts between April and June, depending on latitude. The projected changes in the $SGS_0$ show the largest decrease in mountainous areas, with reductions in the DOY of up to two months. An exception is the Alps, for which the highest situated areas still show a remarkable decrease, but are less dominant than in the Alpine foreland. In terms of low-level regions, especially the continental European and Turkish regions east of 10°E appear striking, with reductions of more than 30 days.

The spatial characteristics of $SGS_5$ are similar to $SGS_0$. However, corresponding to the higher thermal threshold, the occurrence is later within the year. For most regions, the $SGS_5$ occurs around 1.5 to 2.5 months after the $SGS_0$. Within high-elevation areas, the time difference between $SGS_5$ and $SGS_0$ is smaller. The projected changes of the $SGS_5$, however, substantially differ. Throughout the domain, the decreases in the DOY reach 30 to 50 days, with larger decreases in the western parts. While mountainous areas are projected to experience a decrease as well, it is less dominant than in the lower elevated regions. This is particularly true for the Alps.

The spatial characteristics of the $SGS_{10}$ closely resemble those of the $SGS_5$. The shift between the two is around one to 2.5 months and therefore almost linear, when comparing $SGS_0$, $SGS_5$, and $SGS_{10}$. The highest elevated parts of the Alps show no occurrence where nightly temperatures do not remain above 10 °C within the inspected period. In terms of projected changes, the spatial structure of the continental regions resembles that of $SGS_5$, although the decrease is in general on a lower level, reaching from 20 to 30 days. Most noticeable here are most of the coastal regions, where the projected decrease is the highest, with reductions of up to two months.

It becomes apparent from Fig. 3 that the reductions in the DOY for the LDF are disproportional to the reductions in the DOY of the SGS. The level of disproportionality is dependent on the thermal definition of the SGS as well as on regional characteristics such as latitude, altitude, and distance from the sea. From a statistical point of view, only the $SGS_0$ tends to occur prior to the LDF, leading to a potential expectancy that the $FSEI_0$ may be high in the study domain, but not the $FSEI_5$ or the $FSEI_{10}$. However, due to disproportional reductions of the SGS compared to the LDF and with many regions projected to experience a larger reduction in the DOY of the SGS than for the LDF, the risk of an increased FSEI rises.

### 4.2.2 False Spring Event Index (FSEI)

The historical and projected changes of the FSEI are displayed in Fig. 4. In accordance with the assumed distributions described above, the $FSEI_0$ is high in almost all of Southern Europe and Western Asia, as well as in some parts of northwestern Africa, and lies mostly in a range between 15 and 25 (Fig. 4a). By definition, this equals an occurrence ratio of 50-83%. Over the ocean, northeast Africa, and mountainous regions, the FSEI is low. On the contrary, the $FSEI_5$ (Fig. 4e) and $FSEI_{10}$ (Fig. 4i) are in general low within the study domain, with the former reaching around 5 in many parts of the domain and the latter mainly

**Figure 3.** Historical DOY (left) and projected changes in the DOY (right) of the last day of frost (LDF) and the start of the growing season (SGS). First row shows LDF, second row $SGS_0$, third row $SGS_5$, and fourth row $SGS_{10}$.

not occurring over land areas in the historical period. An exception within $FSEI_5$ are the coastal areas of western Europe and

Turkey, where the occurrence is highest at around 10 to 15, equaling 33-50% of all years.

Regarding the projected changes in the future period, Fig. 4b), f), k) display the difference between the decrease in the DOY of the SGS and LDF. In areas marked red, the SGS is projected to retract more quickly towards the beginning of the year

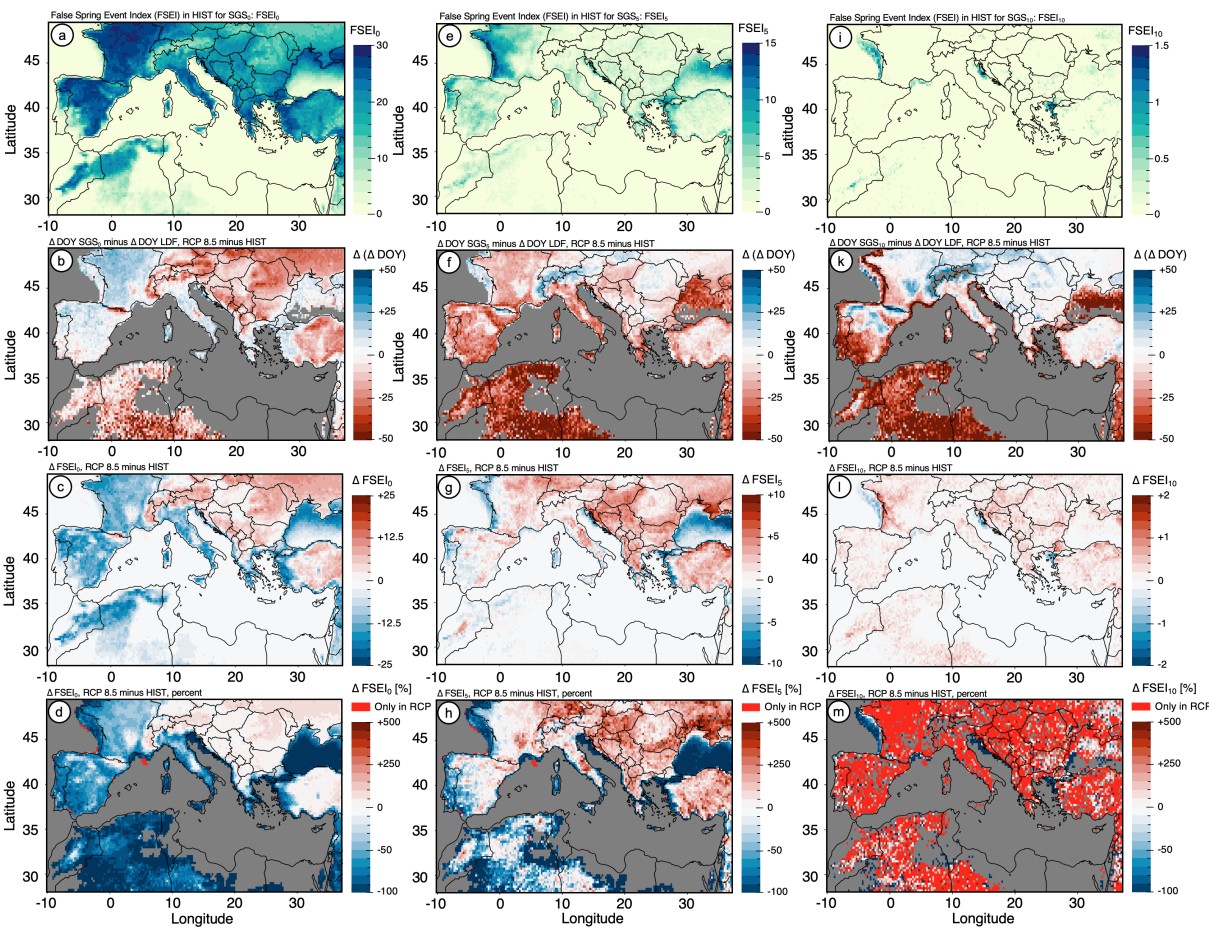

**Figure 4.** Historical occurrence and projected changes of the False Spring Event Index (FSEI). $FSEI_0$ in first column, $FSEI_5$ in second column, $FSEI_{10}$ in third column. Historical occurrence in the first row (a), e), i)), the difference in projected changes in the DOY of LDF and SGS in the second row (b), f), k)), projected changes in the FSEI in the third row (c), g), l)), and projected changes in the FSEI in percent in the fourth row (d), h), m)).

compared to the LDF, therefore potentially increasing the risk of FSEs. The opposite is projected for areas marked blue, i.e. a faster retraction of the LDF compared to the SGS, thus potentially lowering the risk of FSEs. The third row of Fig. 4 shows the

actual projected changes in the $FSEI_0$, $FSEI_5$, and $FSEI_{10}$, and a comparison of the two metrics shows a close agreement for most of the inspected regions. In terms of $FSEI_0$, a decrease of up to 20 events per 30-year period is projected for most of the western and southern parts of the domain. For most of the eastern domain and the mountainous areas except the Atlas, however, up to ten events per 30-year period are projected. For Eastern Europe, a similar picture is apparent for the $FSEI_5$. However, in the western parts of the domain, the signal is reversed and depicts an increase in FSEs of up to 5. Only the most maritime

parts of France, as well as parts of Spain and Portugal, show matching signals for $FSEI_0$ and $FSEI_5$. For $FSEI_{10}$, especially the non-mountainous areas of the study region show a potential increase in the number of events, with mostly one additional





event per 30-year period. Low-level southwestern France, where $FSEI_0$ and $FSEI_5$ and projected to predominantly decrease, becomes most dominant for $FSEI_{10}$, however, with the highest increase of up to 2 events. It should be noted, especially for $FSEI_5$ and $FSEI_{10}$, that some regions appear blue regarding the disproportionate retraction of LDF and SGS, but already show

no occurrence of FSEs in the historical period. Therefore, no further decline in the FSEI can be detected.

The percentage change of the FSEI, shown in the bottom row of Fig. 4, shows a more pronounced manifestation of changes in $FSEI_5$, reaching up to a five-fold increase in the number of FSEs, whereas increases amount to around 50-100% for $FSEI_0$. The decreases projected for $FSEI_0$ amount up to 50-75% of the historical count. While the occurrence of $FSEI_{10}$ is extremely rare in the historical period, Fig. 4 m shows that this event is projected to occur in almost every portion of the study domain in

the future period.

### 4.2.3   Atmospheric variations related to FSEs

The deviation from the mean atmospheric state for sea level pressure (SLP) and daily minimum temperature ($T_n$) for frost events after the $SGS_0$ is shown in Fig. 5 (SLP a), e), h), $T_n$ b), f), i)). Note, that the deviations in the historical period are shown, whereas the deviations in the future period were moved to Appendix B. In general, frost events after SGS are

accompanied by above-average SLP, indicating the potential predominance of high-pressure systems. There is a high level of agreement within the 13 models, as shown by the high number of models indicating this positive SLP deviation (Fig 5e). In addition, these deviations are statistically significant in all 13 models for the majority of areas. A notable exception from this is the high-elevated mountainous regions, where SLP deviations show a negative deviation. As can be expected, $T_n$ deviations are significantly below-average in most low- and mid-elevated areas, where the $SGS_0$ happens particularly early within the

year. In higher elevated areas, where the $SGS_0$ is closer to summer, the deviations turn towards positive values, as frost events occurring post-SGS tend to be warmer than in winter.

In accordance with the climatology of the study domain (see section 2), the wind most often originates from southwesterly to northwesterly directions in the less elevated areas. This changes northerly to easterly directions with decreasing latitude, where the influence of the westerlies decreases. Over the mountain ridges, however, local anomalies appear dominant in the mode

of the wind direction. Here, wind most often blows from the main ridge towards the lowlands, resulting in, e.g., predominant southwesterly winds to the north and east, and northeasterly winds to the south and west of the Apennines. The interplay of large-scale atmospheric flow and local anomalies causes the complex structure shown in Fig. 5c). Under frost events post-SGS, this structure is significantly changed (Fig. 5g), with predominant easterly and northeasterly winds throughout the lowlands (Fig. 5d). Along mountainous areas, the mode of the wind direction appears similar, with winds blowing from the ridge to the

lowlands. However, a high number of models still indicate statistically significant shifts in the wind direction distribution.

The general picture is similar for frost events after the $SGS_5$ (Fig. 6). But with a decreasing number of frost events and therefore a reduced sample size, fewer models indicate statistical significance. For the $SGS_{10}$, with even fewer events, no clear picture can be derived, and the corresponding plot is therefore moved to Appendix B. The overall picture indicates that frost events after the SGS are most often accompanied by cold, easterly airflow under high-pressure conditions, which all

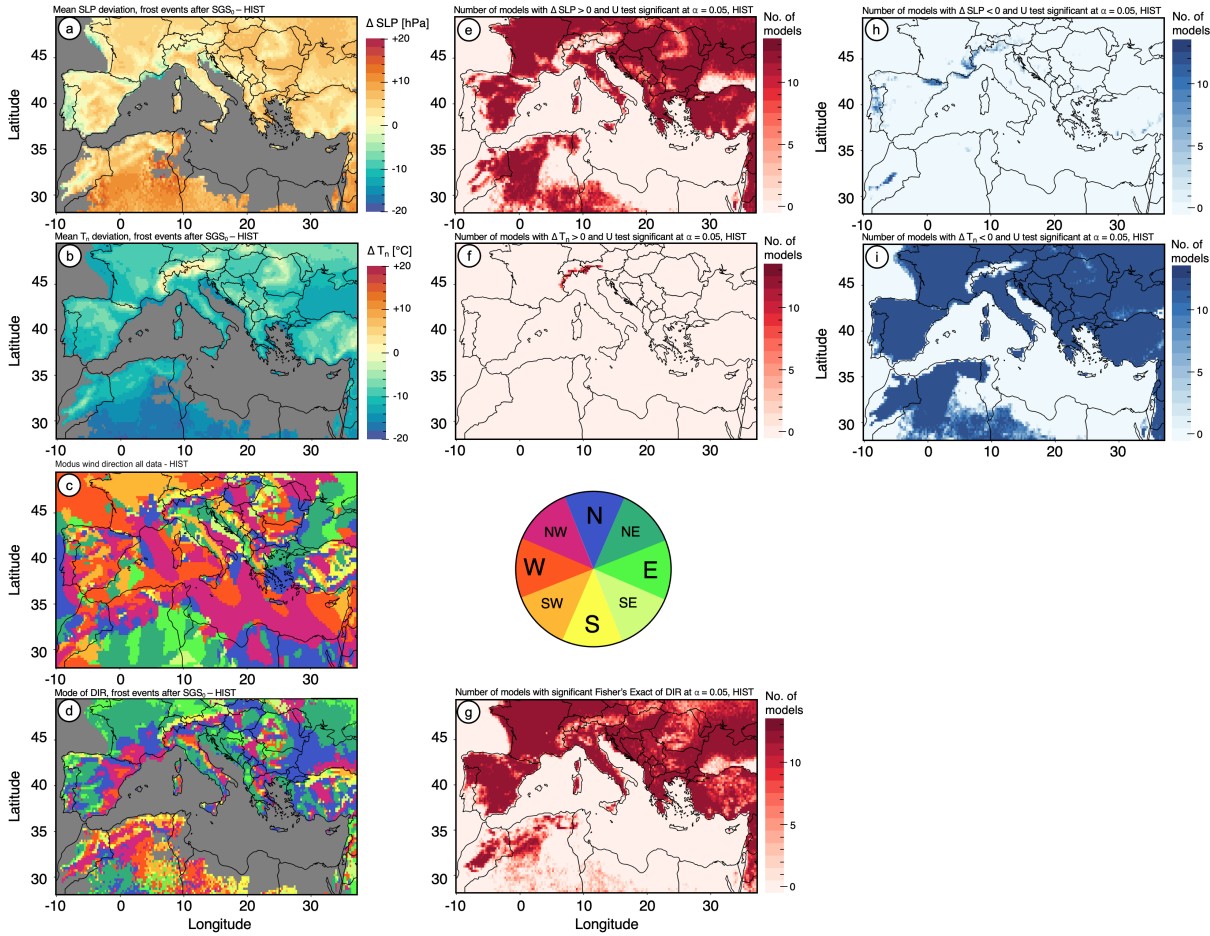

**Figure 5.** Mean deviation of the near-surface atmosphere from the 30-year mean under frost events after the SGS for sea level pressure (SLP, first row) and daily minimum temperature (T$_n$, second row). Number of models indicating a statistically significant positive deviation (e, f), and statistically significant negative deviation (h, i). Mode of wind direction within the 30-year historical period (c) and only for frost events after the SGS (d). Number of models indicating significantly different distributions of wind directions (g).

significantly deviate from the 30-year mean. In mountainous regions, descending air masses flowing from the ridge to the lowlands are predominant.

In the future, there is no indication of a change to the general picture of these characteristics. While cold, easterly flows under high pressure are also predominant in the projections, fewer models indicate a statistically significant deviation from the 30-year mean (Appendix B). Comparing the mean atmospheric deviation in the future and historical periods, it can be seen in

Fig. 7 that the models project an increase in the positive deviation of SLP from the mean state for most of Eastern Europe and Western Asia.
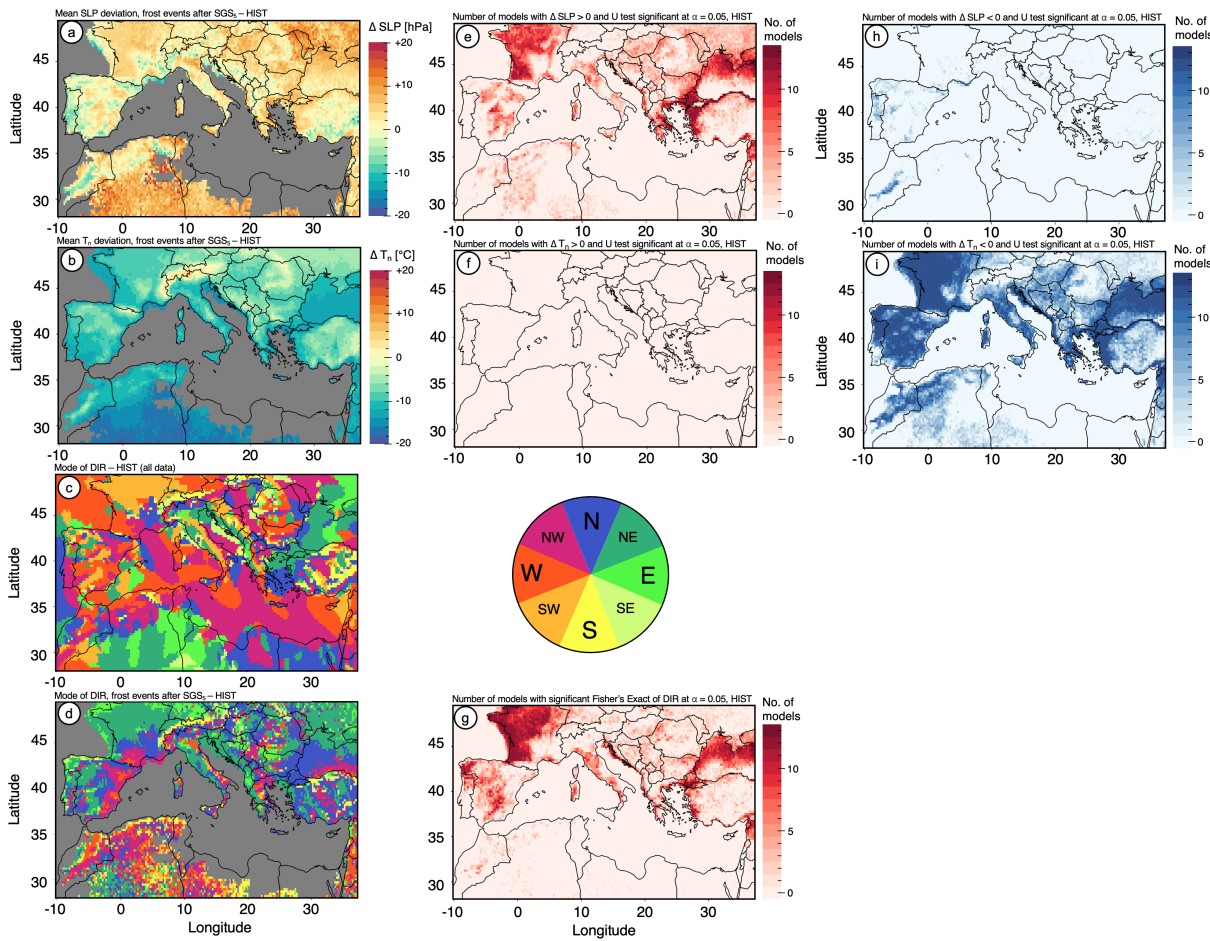

**Figure 6.** Same as Fig. 5, but for FSEI$_5$.

The homogeneous spatial indication of this change for SGS$_0$ is less uniform for SGS$_5$, where the majority of cells indicate an increase in positive deviation mixed with cells indicating a decrease. In Spain, Portugal, and northwestern Africa, however, the projected development is the opposite, indicating a general decrease in positive SLP deviation. In terms of T$_n$, regions with varying changes in the deviation mix closely for SGS$_0$ with no clear spatial distinction. For SGS$_5$, which is in general projected to occur earlier within a year, the temperature deviation is projected to decrease in correspondence with a higher potential for lower temperatures closer to the winter period. In general, not accounting for regional variations, the projections indicate an intensification of the high-pressure anomaly under frost events post-SGS, with a higher potential for lower temperature anomalies than in the historical period.


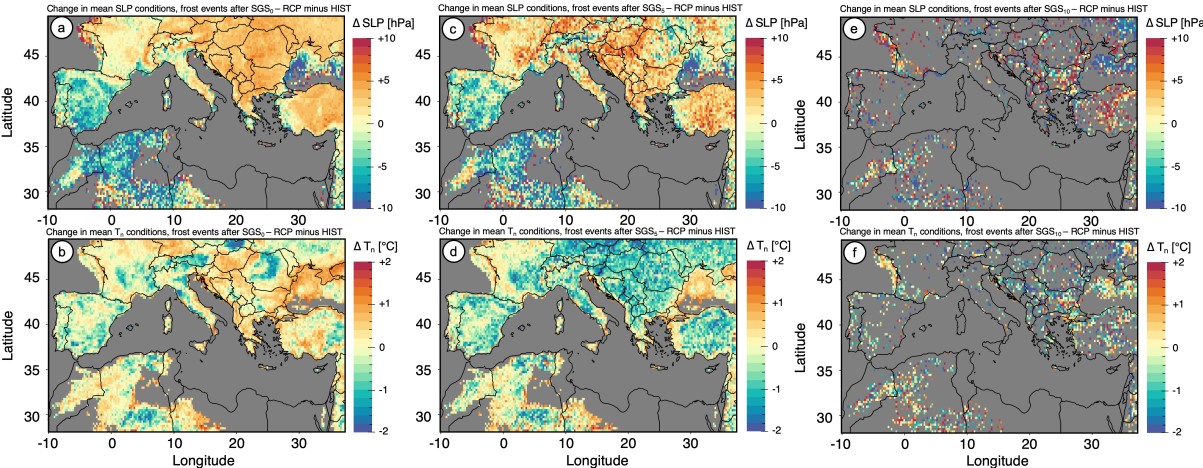

**Figure 7.** Projected changes in the deviations of sea level pressure (SLP, first row) and daily minimum temperature ($T_n$, second row) from the 30-year mean under frost events after the SGS. $SGS_0$ (a-b), $SGS_5$ (c-d), and $SGS_{10}$ (e-f).

## 4.3 Compound events of heat and drought

### 4.3.1 SPI-3 and percentiles of $T_x$

Prerequisites for the investigation of heat and drought-related compound events are changes in the percentiles of daily maximum temperature ($T_x$) and the corresponding change in the exceedance of these thresholds, as well as changes in drought behavior indicated by the SPI-3. These projected changes are shown in Fig. 8.

In accordance with the climatological description in section 2, the thermal gradients regarding latitude, elevation, and distance from the sea become apparent for the historical percentile-based thresholds of $T_x$ including the $90^{th}$, $95^{th}$, and $99^{th}$ percentiles in Fig. 8a) and c). The given values all refer to the meteorological summer, including June, July, and August (JJA). The corresponding projected changes under RCP 8.5 are shown in Fig. 8d) and f) and depict a minimum increase of 4.5-5 °C for a majority of land areas in the study domain. Additionally, a southward increase in the deviation can be detected, reaching from around +5 °C in central Europe to over +6 °C in northwestern Africa. Thirdly, some mountainous areas, e.g. the Alps and the Pyrenees, show intensified warming compared to the surrounding lowlands. Next to the spatial gradients, an increase in warming can be detected for increasing percentiles, which is true for most parts of the study domain. In general, when applying the percentile-based thresholds derived from the historical period (see. Fig. 2c) to the future daily $T_x$, the historical $90^{th}$ percentile is projected to shift to approximately the $66^{th}$ percentile. Therefore, while this threshold was historically exceeded on one of ten days on average, one out of three days would exceed the historical threshold in the future. This tripling of the occurrence probability is surpassed by the $99^{th}$ percentile, for which a fifteen-fold increase is projected.

The SPI-3 indicates an increased probability for drought conditions in the study domain, excluding parts of central and eastern Europe, which is reflected both in the annual (Fig. 8g) and JJA consideration (Fig. 8h). However, lower values for JJA

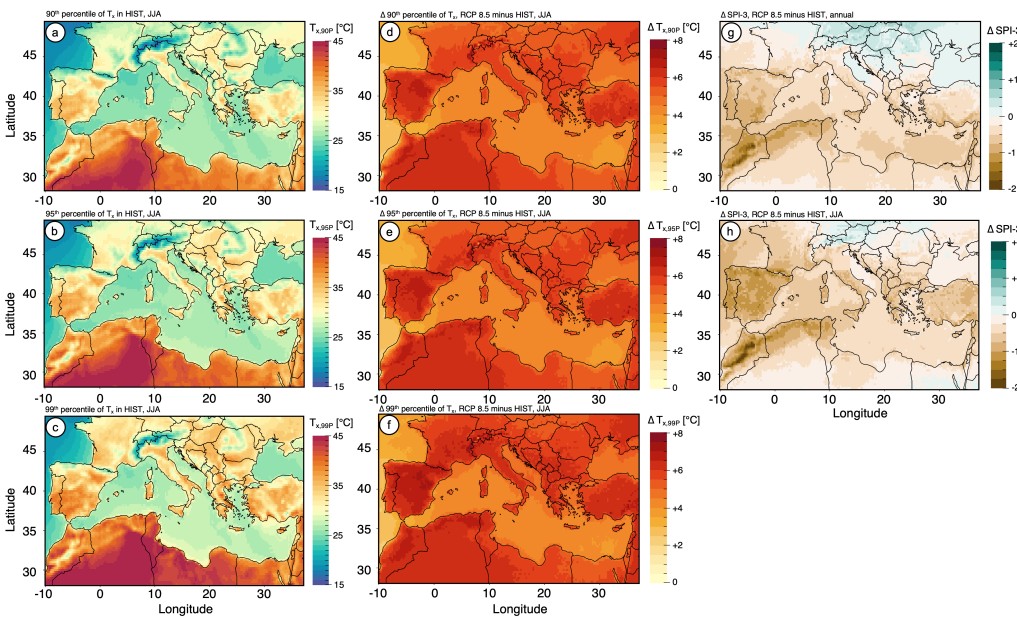

**Figure 8.** Historical (left column) and projected changes (middle column) in the $90^{th}$ (a), d)), $95^{th}$ (b), e)), and $99^{th}$ (c), f)) percentile of daily maximum temperature. Projected deviation of the SPI-3 from the historical mean in the future period (right column) under annual (g) and summer (h) consideration.

indicate an aggravated drought situation for the summer months, compared to the annual mean. For most regions indicating

a decrease in the SPI-3, the reductions are projected at around 0.5. This equals a reduction in the mean precipitation sum of around one-half standard deviation of the historical period. However, in the western parts of the domain, including most parts of Spain, Portugal, and Morocco, the SPI-3 is projected to decrease by more than one standard deviation, indicating prevailing drought conditions in the future. In the regions where $\Delta$SPI-3 falls below -1, on average, every month is expected to meet the drought-based requisite for the occurrence of HDCEs.

**4.3.2 Heat-Drought Compound Events (HDCEs)**

The historical and future spatial characteristics of HDCEs, as well as the corresponding length of consecutive days meeting the HW criteria, are shown in Fig. 9. During the historical period (Fig. 9a), e), i)), $HDCE_{90}$ most often occurred in the northern half of the domain, including Italy, southern France, and the Balkans, reaching occurrence rates of once every three to five years, on average. While the spatial differences are similar for $HDCE_{95}$, the frequency of events is reduced, mostly occurring

less than once every ten years. $HDCE_{99}$ are virtually not simulated by the models, showing only singular appearances within the 30-year period. Due to the low historical occurrence rates compared to the future period, the absolute occurrences, instead of the $\Delta$, are shown in Fig. 9b), f), k). For every grid cell of the domain, increases in the occurrence of all types of HDCEs are projected. The largest frequency over land areas appears in regions close to, or bordering, the Mediterranean Sea. For $HDCE_{90}$,



up to 20–25 days per summer are projected to occur in these regions, with frequencies dropping to around 5–10 events per

year in Central Europe and northeastern Africa. As for the historical occurrence, the spatial structure of the projected increases is similar for HDCE$_{95}$ and HDCE$_{99}$, with decreasing frequencies for higher thermal thresholds. Within the aforementioned regions with the highest increase, HDCE$_{95}$ are projected to occur around 10–20 times per summer, and 5–10 times per summer for HDCE$_{99}$.

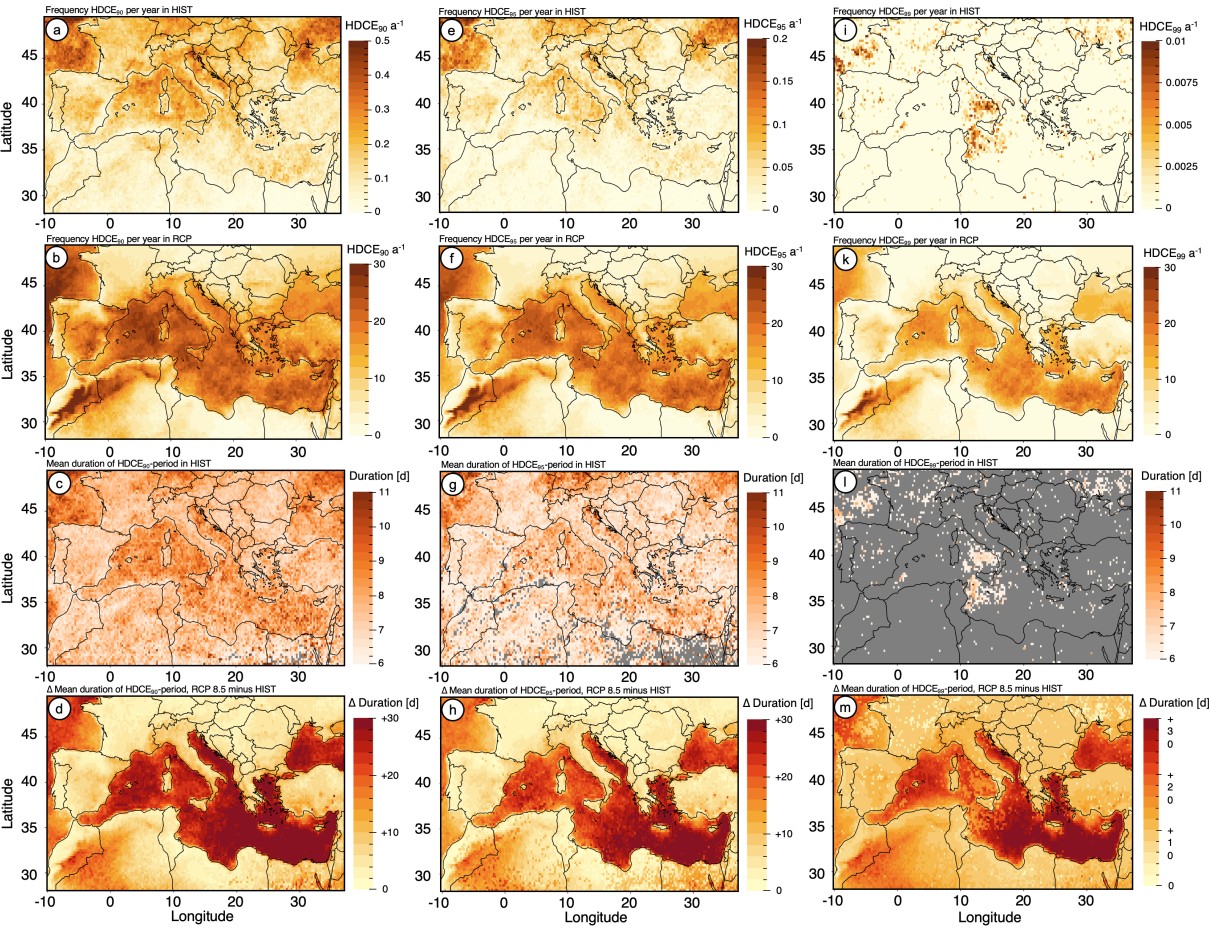

**Figure 9.** Historical occurrence and projected changes of Heat-Drought Compound Events (HDCEs). HDCE$_{90}$ in the first column, HDCE$_{95}$ in the second column, HDCE$_{99}$ in the third column. Historical occurrence rate in the first row (a), e), i)), projected future occurrence rate in the second row (b), f), k)). Average historical length of consecutive days meeting the HW criteria (HDCE-period) in the third row (c), g), l)) and projected changes in the average length of HDCE-periods in the fourth row (d), h), m)).

In accordance with the definition of HDCEs, the number of consecutive days meeting the HW criteria (HDCE-period) is

at least six. In the historical period, the average HDCE-period has a length of around seven to nine days for HDCE$_{90}$, with no apparent spatial discrimination (Fig. 9c). The length is reduced to around seven to eight days for HDCE$_{95}$. The projected





changes of the HDCE-period are highest in the regions with the highest projected increase in HDCEs (Fig. 9d), h), m)). For HDCE$_{90}$, gains reach up to 15 days, resulting in an overall projected length of around three weeks, on average. Increases in the length of the HDCE-period reach 10 days for HDCE$_{95}$, resulting in more than two weeks with consecutive days reaching the

HDCE$_{95}$ criteria. Most land areas show an increase of six days for HDCE$_{99}$, which is mostly due to HDCE$_{99}$ being projected for the first time within the future period.

### 4.3.3    Atmospheric variations related to HDCEs

For this investigation, all days meeting the HDCE criteria were extracted, and their mean climatology was compared to that of the entire 30-year period. Due to the low number of events (see Fig. 9) the results on near-surface atmospheric variations during

the historical period were moved to the appendix. The general picture of the variations detected in the future period similarly applies to the historical period (comp. Fig. 10-12 and Appendix B). As shown by Fig. 10, HDCE$_{90}$ are accompanied by below-average SLP conditions over most of the low elevated areas. These negative deviations are often statistically significant, although the number of models varies.

     Over mountainous regions, however, the SLP deviations are positive, indicating above-average pressure conditions. Over

the Alps, Pyrenees, and Atlas, these deviations are statistically significant in a majority of models. As expectable, the T$_x$ deviations are significantly above-average. The mode of the wind direction is similar for both the entire 30-year period and for HDCEs. As for FSEs, westerly to northerly wind directions are predominant in the eastern half of the study domain. In the low elevated areas of the northwestern parts of the domain, where westerly winds are predominant for FSEs (see Fig. 5), winds predominantly originate from the north and northeast. Mountainous areas, similar to FSEs, are determined by regional

characteristics depending on the orientation of the main ridge, causing a complex picture in the mode of the wind direction. Contrary to FSEs, near-surface flows tend to move towards the mountain ridge, as can be seen for the Atlas, Pyrenees, and the Apennines (Fig. 10c). For many land areas in the study domain, the projected distribution of the wind directions under HDCE$_{90}$ in the future differs significantly from the total distribution (Fig. 10g).

     The overall picture of HDCE$_{95}$ and HDCE$_{99}$ is similar to HDCE$_{90}$, with below-average sea level pressure conditions in the

low-lying areas and above-average conditions over the western mountain ridges (HDCE$_{95}$ in Fig. 11, HDCE$_{99}$ in Appendix C). Winds predominantly originate from northern to eastern directions, excluding regional phenomena in the vicinity of mountain ranges. While a majority of the 13 inspected models prove the atmospheric variations under HDCEs to differ from the overall mean with statistical significance, the number of indicating models decreases with decreasing sample size, resp. higher thermal thresholds.

The projected changes in the mean near-atmospheric state under HDCEs are shown in Fig. 12. The results described above for the future period are therefore the result of a decrease in sea level pressure for most regions. Still, some regions show an increase in sea level pressure, adding to the complexity of the projected changes. The deviation in mean T$_x$ under HDCEs, as indicated by the general projected warming, is shown to increase by around 2.5-5 °C, and often the projected warming is more intense for HDCE$_{95}$ than for HDCE$_{90}$. Due to the low occurrence rate in the historical period, comparisons for HDCE$_{99}$ could

not be made.


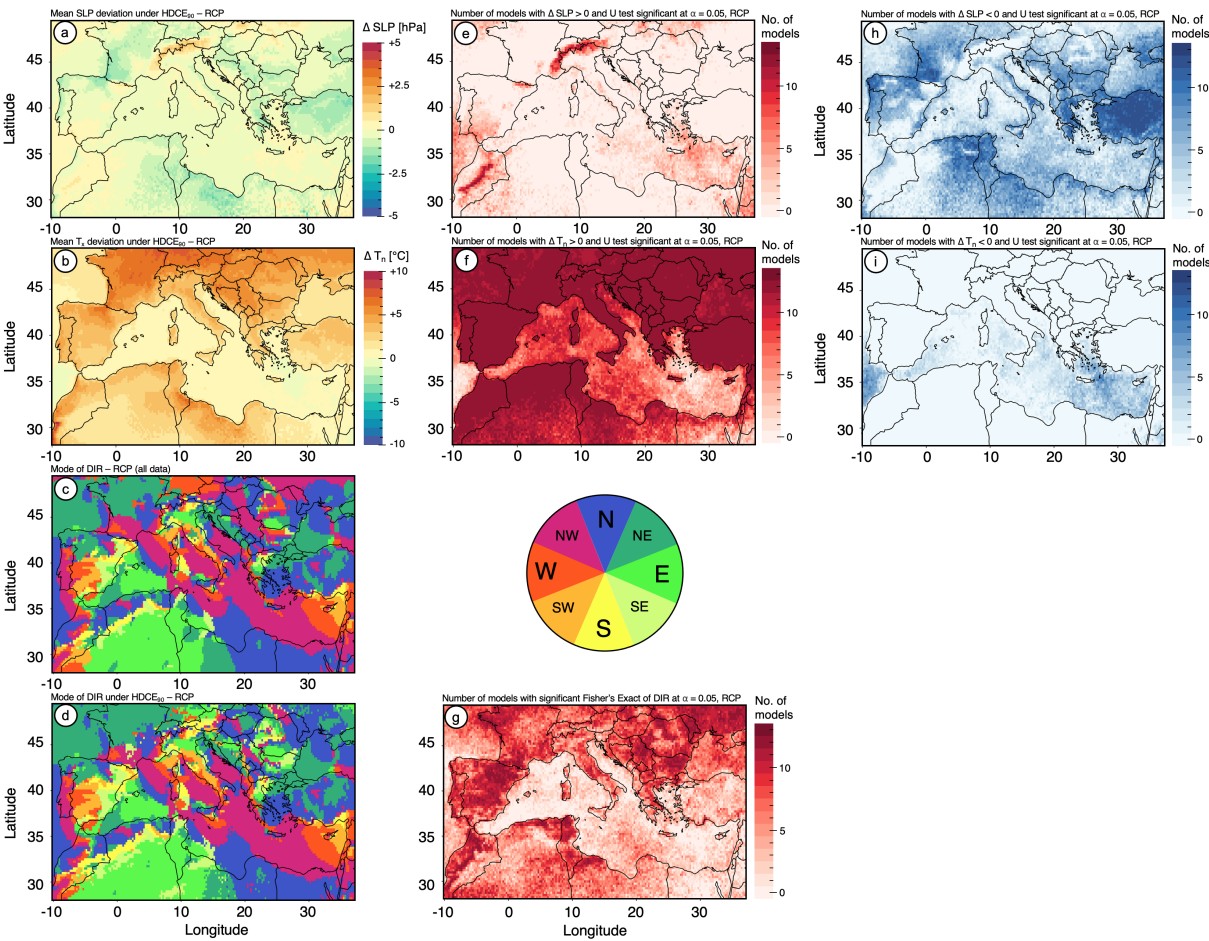

**Figure 10.** Mean deviation of the near-surface atmosphere from the 30-year mean under $HDCE_{90}$ for sea level pressure (SLP, first row) and daily maximum temperature ($T_x$, second row). Number of models indicating a statistically significant positive deviation (e, f), and statistically significant negative deviation (h, i). Mode of wind direction within the 30-year historical period (c) and only for HDCEs (d). Number of models indicating significantly differing distributions of wind directions (g).

## 5   Discussion

The use of dynamically downscaled climate model output offers several advantages over the output of General Circulation Models, foremost a higher spatial and temporal resolution, as well as a higher data accuracy. However, as the results of this study prove for the GMR, a high amount of bias remains in the output of RCMs, when compared to state-of-the-art reanalysis data.
This bias varies for different variables and is dependent on the underlying GCM and RCM, but it is not clearly distinguishable whether the choice of GCM or RCM contributes more bias. With one exception, temperatures are underestimated in the raw model output, which is in line with former experiences with CORDEX output (Dosio et al., 2022; Top et al., 2021). Precipitation sums, on the other hand, show an almost equal number of over- and underestimating models. However, the magnitude of the bias

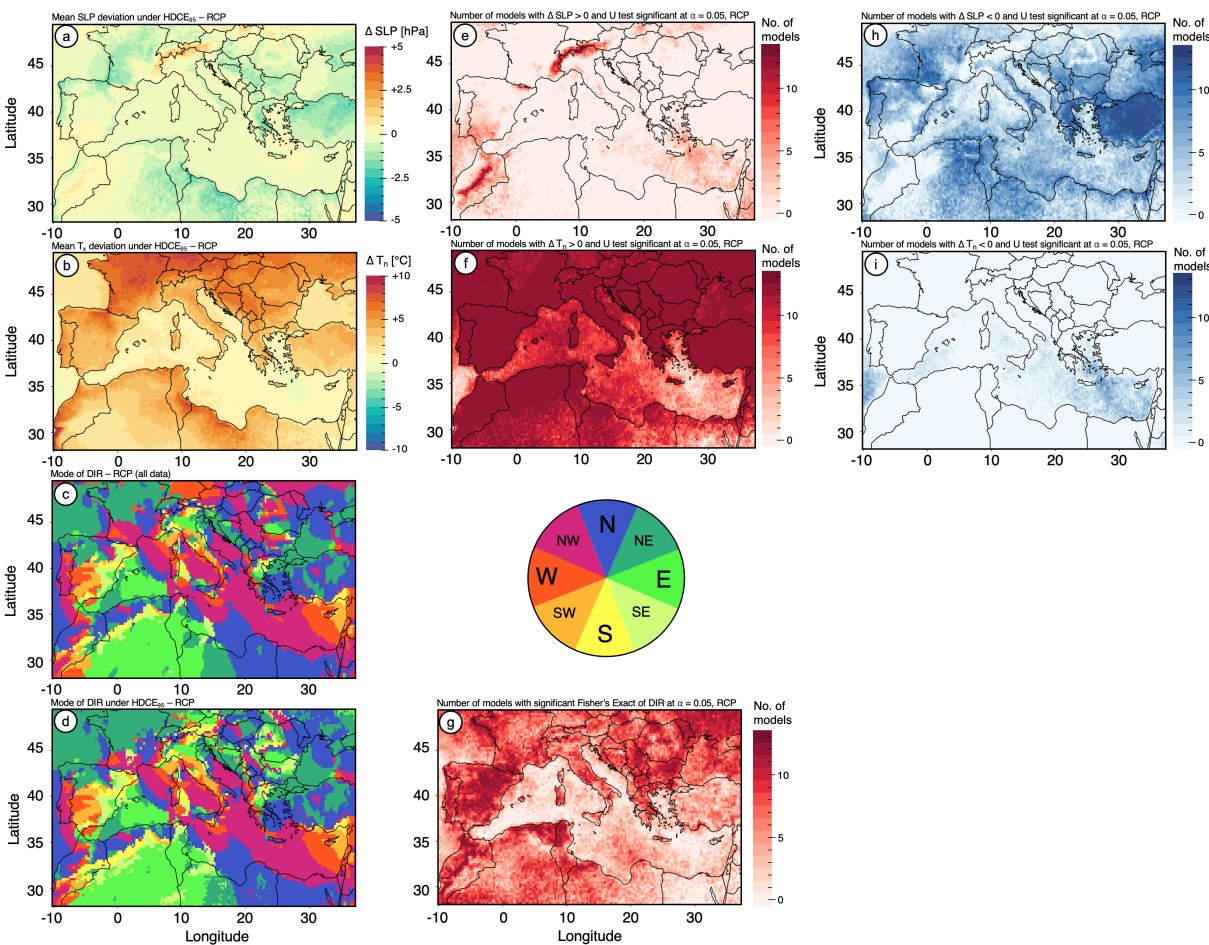

**Figure 11.** Same as Fig. 9, but for HDCE$_{95}$.

is higher for models with an overestimation of the precipitation sum. Reasons for the over- and underestimation of precipitation,
including the "drizzle-effect", have been discussed before, e.g. by Chen et al. (2021), Demory et al. (2020), and DeMott et al.
(2007). A bias is present in both the climatology of daily data, i.e. percentiles, and the long-term climatology. After applying
QDM, both mentioned forms of bias are removed to a significant degree, resulting in nearly zero bias in terms of percentiles
and minimal deviations in long-term climatology, depending on the selected GCM-RCM combination and variable. Therefore,
and by preserving the dependence structure of multiple corrected variables, MBCn proves to be a suitable choice for assessing
percentile threshold-based compound events. For this study, the main focus was to explore vegetation-relevant compound
events and discuss the potential associated risks that could unfold in the future, after applying a multivariate bias correction
method that is proven to deliver reliable results. It was therefore not the aim of this study to assess the opportunities and
limitations of different methods. For example, the performance of several different multivariate bias correction methods could
be evaluated in follow-up studies, similar to what has been done before for univariate correction methods applied to CORDEX





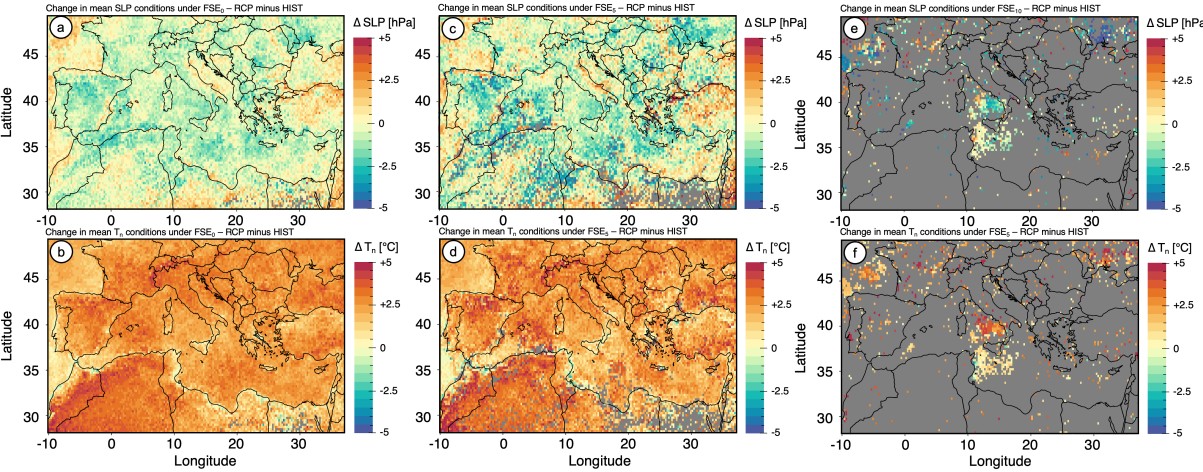

**Figure 12.** Projected changes in the deviations of sea level pressure (SLP, first row) and daily maximum temperature ($T_x$, second row) from the 30-year mean under HDCEs. $HDCE_{90}$ (a-b), $HDCE_{95}$ (c-d), and $HDCE_{99}$ (e-f).

(Olschewski et al., 2023; Laux et al., 2021). An additional limitation of this study lies in the choice of reference data. While ERA5 is proven to be a suitable state-of-the-art reference for temperature and precipitation (Lavers et al., 2022; Mistry et al., 2022; Velikou et al., 2022; Hassler and Lauer, 2021), some level of uncertainty regarding extreme events remains, especially for precipitation. When conducting local refinement of the presented results in potential follow-up studies, additional reference data sets, including station-based observations, could be compared to further improve the robustness of the projections.

The disproportionate behavior of the LDF and the SGS shown in this study has also been demonstrated, for example, for the United States (Allstadt et al., 2015; Peterson and Abatzoglou, 2014; Marino et al., 2011). These authors come to the conclusion that FSE-related risk varies significantly for different regions. In this context, Chamberlain et al. (2021) discuss potential reasons for the large regional variability, including elevation (Vitasse et al., 2018) and distance from the sea (Ma et al., 2019). The findings of this study also demonstrate these two drivers as decisive when determining the potential future risk of FSEs.

However, the nature of the projected change is also strongly dependent on the selected SGS threshold. Regarding the SGS, the presented thresholds are strictly based on atmospheric temperatures and do not include data on, e.g., soil temperatures and observations of vegetation. Apart from highly elevated regions, where the onset of the growing season is generally later in the year, the three selected thermal thresholds for the SGS are proven in this study to represent three different sections within the warming period between January and June. This offers a benefit for a future refinement of the presented results, for example for

specific crops with specific thermal characteristics referring to one of the sections. The broad nature of the thermal definitions yet imposes a limitation on the results of this study. A more sophisticated approach, e.g. including crop-related modeling or crop-specific thresholds of budburst and freezing damage, for example, discussed by Allstadt et al. (2015) and Chamberlain et al. (2019), may lead to an increased preciseness and applicability for practitioners. On the other hand, the aim of this study is to provide a general overview of the potential risk of FSEs within a large domain that has not been closely investigated yet in





this concern. This was done, using simplified and straightforward thermal definitions. This study proves the spatial resolution to be sufficient to distinguish regional characteristics of FSEs. In terms of agricultural applications on a local level, where crop-specific metrics and local climate characteristics gain in importance, the presented results may be the starting point of local refinement.

The results regarding the atmospheric deviations under FSEs are in-line with the general understanding of climatology in
the study domain, and this especially applies to the low-lying areas. During spring, air masses originating from westerly and southerly directions are more and more likely to cause warm conditions, as the level of warming closely follows the northward shift of the subtropical ridge (see section 2). However, especially when high-pressure conditions cause clear skies and the nights are still long, a high level of outgoing longwave radiation over land masses causes temperature levels to decrease. Under easterly flows, which this study proves to be predominant under FSEs, these cooled air masses are forced into the study
domain and increase the risk of frost events. In a recent study for Europe, Quesada et al. (2023) determined the advection of northerly to easterly air masses to be crucial drivers of cold waves over Europe. These results can be directly linked to the findings of this study, increasing the predictability of FSEs. Over mountainous areas, the predominant winds differ, indicating flows from the mountain ridges towards the lowland. In general, descending air masses are linked to dissolving cloud coverage, resulting in clear skies. As described above, the longwave outgoing radiation can therefore not be retained in the lower
atmosphere, contributing to low near-surface temperatures and the occurrence of FSEs. On the other hand, for example, under foehn conditions, descending air masses on the leeward side of a mountain ridge may be linked to warm conditions (Jansing et al., 2022) and therefore contradict the results of this study. To gain further insights into the local conditions caused by the large-scale atmospheric steer flow, additional vertical layers of atmospheric flow that are less sensitive to local orography, e.g. the 500 hPa layer, could be assessed in further studies. For this study, we specifically focused on the near-surface atmospheric
conditions, as these are most relevant in the context of crops. A similar gain may also be achieved by considering circulation patterns, for example, based on Jenkinson and Collison (1977) or Hess and Brezowsky (James, 2007). However, the application of large-scale circulation patterns is aggravated in the context of multivariate bias correction, as the spatial coherence, next to the temporal coherence, of all variables must be preserved for the entire study domain. As Cannon (2018) pointed out, this is, in general, possible for MBCn. However, considering the size of the study domain, the additional consideration of spatial
coherence required a significantly higher computational effort that was not available within the means of this study.

The nature of compound extreme events including heat and drought over Europe and the GMR has been extensively studied for historical periods (Vogel et al., 2021; Ionita et al., 2021; De Luca et al., 2020; Russo et al., 2019). The results presented in this study embody the continuation of the historical trend of increasing HDCEs that all former studies share. Inspecting the relevant variables independently already shows the tendency of each variable to favor the occurrence of HDCEs in the future.
As described above, the projected increase in daily maximum temperature for the spatial mean causes extremes exceeding the $99^{th}$ percentile to be 15 times more likely. However, as a majority of regions show the potential for persistent drought during summer (JJA), the projected change in the occurrence rate of HDCEs is multiple times higher. This aggravated risk of compounding extremes has, for example, been indicated by Zscheischler et al. (2018). The potential effects of the projected changes in heat and drought behavior are diverse. An increased risk of crop failure, for example, has been shown by He et al.




(2022) on a global level and by Ribeiro et al. (2020) for a Mediterranean domain. While it must be noted, that crop failure risk is highly dependent on the crop type and local climate factors, accumulated drought and heat nevertheless have negative impacts on the amount of water available for irrigation. Apart from agricultural aspects and natural ecosystems, the projected level of warming will significantly impact human health and mortality (Raymond et al., 2020; Gasparrini et al., 2015). The impact of increased temperatures is also shown to be aggravated in highly urbanized areas (Merkenschlager et al., 2023) and

further amplifies socioeconomic disparities (Hsu et al., 2021).

The results of this study depict that below-average sea level pressure conditions and predominantly northeasterly to easterly flows are accompanied by HDCEs. A reason for below-average pressure could be near-surface thermal lows induced by intense surface warming, as previously described by Lavaysse et al. (2016) and Hoinka and Castro (2003). This, in combination with the advection of dry, continental air masses from eastern Europe, may be crucial for the occurrence of HDCEs. The limitations

in terms of drawing conclusions on the large-scale atmospheric circulation, as discussed above, similarly apply to HDCEs. While the results clearly depict significantly differing conditions under HDCEs and offer benefits regarding the predictability of these events, a follow-up study including circulations patterns could improve the quality of these predictions even more. This also applies to the mountainous regions, which show different characteristics under HDCEs than the surrounding lowlands. For both, FSEs and HDCEs, the climate projections show an intensification of the pressure anomalies. For FSEs, the already above-

average conditions show a projected increase in deviation from the current mean atmospheric state. For HDCEs, there is a less clear picture, although more regions show below-average conditions for historical HDCEs. This below-average deviation is amplified in the future projections for a majority of regions in the study domain. Whether centers of action (Osman et al., 2021) and related blocking systems will intensify in the future is subject to the scientific discourse, as discussed by Kautz et al. (2022).

## 6 Summary and conclusions

The aim of this study is to demonstrate characteristics and potential changes in False Spring Events (FSEs) and Heat-Drought Compound Events (HDCEs) in a high-impact future scenario (RCP8.5) for the end of the $21^{st}$ century. The inspected periods are 1970-1999 and 2070-2099. We applied a multivariate, i.e., a variable dependence-preserving, bias correction method based on the N-dimensional probability density function transform (MBCn) to regional climate model output obtained from

CORDEX. The results prove that MBCn can successfully remove biases in the model output and therefore increase the reliability of projections. Due to MBCn representing a percentile-adjusting method, this is specifically true for the percentile distributions of jointly corrected climate variables, making it a valuable method for the derivation and inspection of percentile threshold-based compound extreme events.

The presented results indicate that, while only FSEs of the lowest thermal threshold ($SGS_0$) are historically relevant on a

widespread basis in the Greater Mediterranean Region, FSEs are projected to gain in frequency and relevance in the future period. This is mainly due to a disproportionate reduction in the day of the year (DOY) of the last day of frost (LDF) and the start of the growing season (SGS). Therefore, warm periods have an increased probability of occurring particularly early in the





year, whereas the risk of frost events remains. In low-lying areas, FSEs are mostly coupled with high-pressure anomalies and northerly to easterly winds. This is in line with cold, continental air masses flowing into the GMR and causing an increased
risk of frost events. For mountainous areas, local winds appear predominant, mostly flowing from the mountain ridge towards the lowlands.

All investigated metrics of HDCEs indicate significant changes in the future. These include an increase in intensity, with more severe phases of agricultural drought and increased daily maximum temperatures of up to 6 °C. Also, the frequency of HDCEs and the length of consecutive days meeting the HDCE criteria is projected to increase. In general, the changes in
HDCEs are multiple times more intense than those of the underlying univariate extremes. Local thermal lows are shown as the predominant near-surface condition linked to the occurrence of HDCEs, together with the inflow of dry, continental air masses from northern to eastern directions.

The projected changes will potentially have severe negative effects on vegetation and crop efficiency if no additional adaptive measures are applied. An increased risk of frost exposure due to an earlier SGS can have adverse effects on crop yield by
causing damage in the early stages of development. This is particularly true for plant species with only rare historical exposure, and therefore a low degree of adaption to frost. The projected level of heat and drought-induced stress could potentially aggravate the agricultural and socioeconomic stress even more. This could, for example, manifest itself in the form of severe water shortages for humans, livestock, and vegetation, stress on food security due to the increased risk of crop losses or failures, increased tree and forest mortality endangering natural ecosystems, and deteriorated human health and increased mortality.

While the public focus often lies on the effects of increased temperature levels due to global warming, this study demonstrates the additional need for adaptation on the other end of the thermal spectrum. To prevent vegetation and crop damage or loss, which would further amplify the already aggravated ecological and nutritional stress, the efforts to protect vegetation and crops should equally focus on heat/drought and frost. The results of this study can offer benefits in this regard to allow for better-informed adaptation strategies in agriculture. An additional benefit is given by increasing the level of predictability of FSEs
and HDCEs. This will allow local actors to react to forecasts indicating a potential occurrence in a timely manner. The data sets generated in this study can additionally be used for further climate change impact analyses, that, for example, focus on a local refinement or specific crop species in the GMR. By applying more complex definitions of compound extreme events, for example, when additional predictors or higher resolution data sets are available, potential follow-up studies will be able to further improve the assessment capability of compound extreme events.

*Author contributions.* P.O., B.B., and P.L. conceptualized the study design and aims. P.O. and E.H. downloaded and curated the data. P.O. and D.D. wrote and implemented the code. P.O. carried out the calculations. P.O., D.D., H.M., B.B., and P.L. evaluated and interpreted the results. H.K. and P.L. acted as supervisors. P.O. wrote and prepared the draft manuscript. All authors reviewed and approved the final version of the manuscript.



*Competing interests.* The authors declare that they have no known competing financial interests or personal relationships that could have appeared to influence the work reported in this paper.

*Financial support.* This research did not receive any specific grant from funding agencies in the public, commercial, or not-for-profit sectors.

*Acknowledgements.* The computations for this analysis were carried out using the programming environment R, version 4.2.2 (R Core Team, 2022). NetCDF data was processed within the R framework "climate4R", provided by Santander Meteorology Group (Iturbide et al., 2018). Bias correction was performed using the "MBC" package (Cannon, 2023, 2018; Cannon et al., 2015). We thankfully acknowledge the public provision of the CORDEX (Cinquini et al., 2014) and ERA5 (Hersbach et al., 2023) data. The study has been performed in the wider context of the research project *BENEFIT-Med: Boosting technologies of orphan legumes towards resilient farming systems in the Greater Mediterranean Region: from bench to open field* (without dedicated funding). Ideas from the project team discussions partly contributed to the conceptualization of this study.

*Data availability.* All data sets that were obtained for this study are openly accessible. The ERA5 data can be downloaded from the Copernicus Climate Change Service (C3S) Climate Data Store at https://cds.climate.copernicus.eu/cdsapp#!/dataset/reanalysis-era5-single-levels?tab=overview, the CORDEX data can be downloaded, for example, via the DKRZ node of the ESGF data portal at https://esgf-data.dkrz.de/search/cordex-dkrz/. Both are retrievable upon registration.

## Appendix A: Supplementary material for Bias correction performance



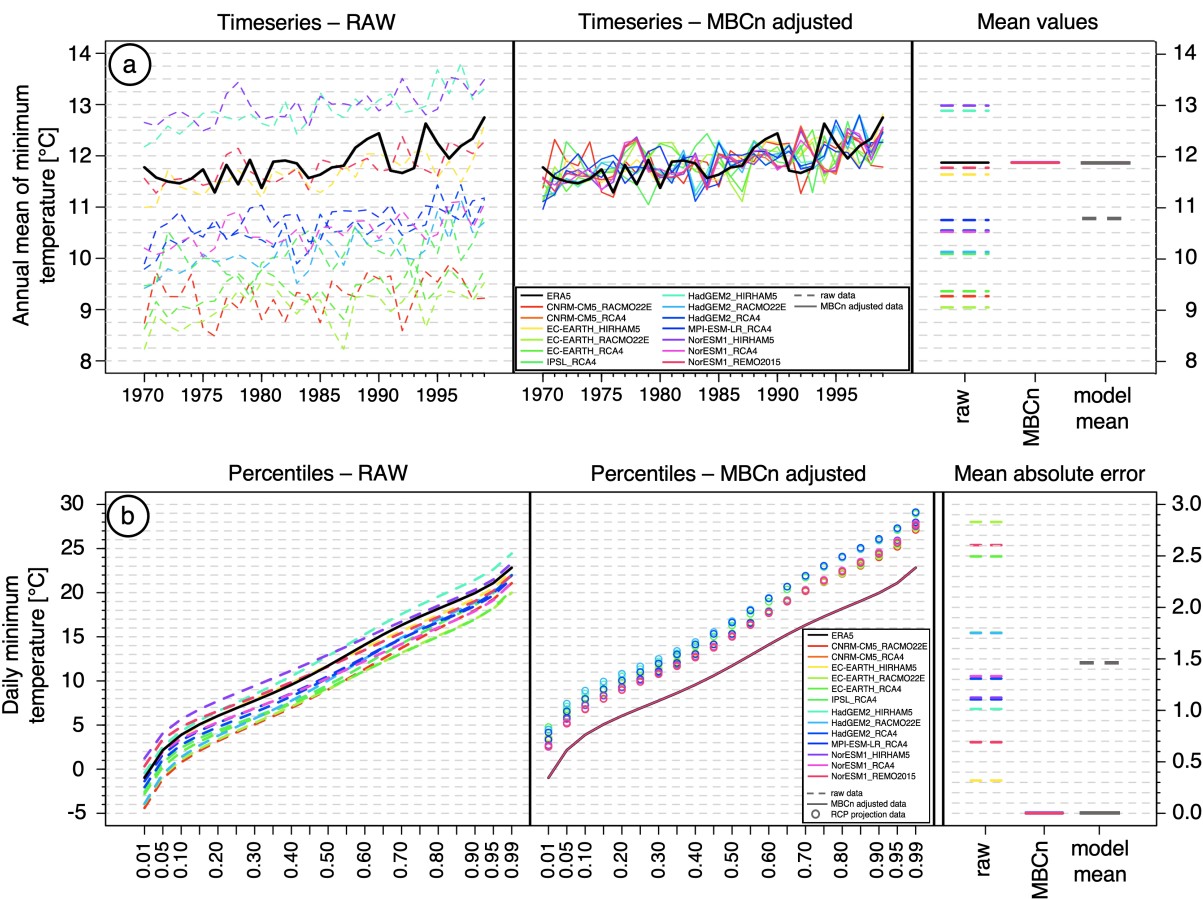

**Figure A1.** Evaluation of MBCn performance for long-term climatology and daily-based percentiles. a) Long-term time series of annual mean minimum temperature for ERA5 and 13 CORDEX models in historical raw (left), historical MBCn-corrected output (middle), and 30-year mean values (right). b) percentile-based distributions of daily minimum temperature for ERA5 and 13 CORDEX models in historical raw data (left), MBCn-corrected historical and projected output (middle), and mean absolute error (right).



## Appendix B: Supplementary material for FSEs

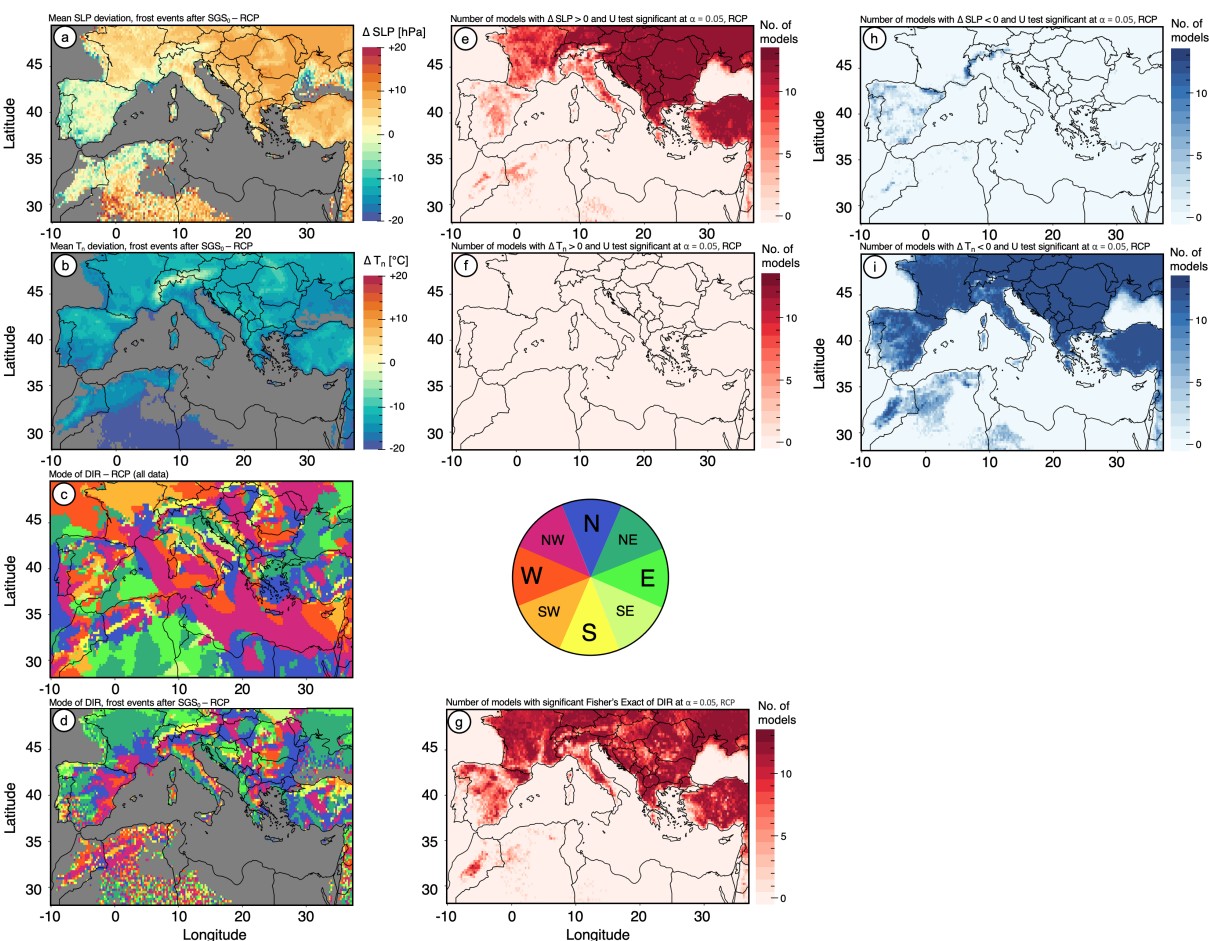

**Figure B1.** Mean deviation of the near-surface atmosphere from the 30-year mean under frost events after the $SGS_0$ for sea level pressure (SLP, first row) and daily minimum temperature ($T_n$, second row). Future deviations under RCP8.5. Number of models indicating a statistically significant positive deviation (e, f), and statistically significant negative deviation (h, i). Mode of wind direction within the 30-year future period (c) and only for frost events after the $SGS_0$ (d). Number of models indicating significantly differing distributions of wind directions (g).

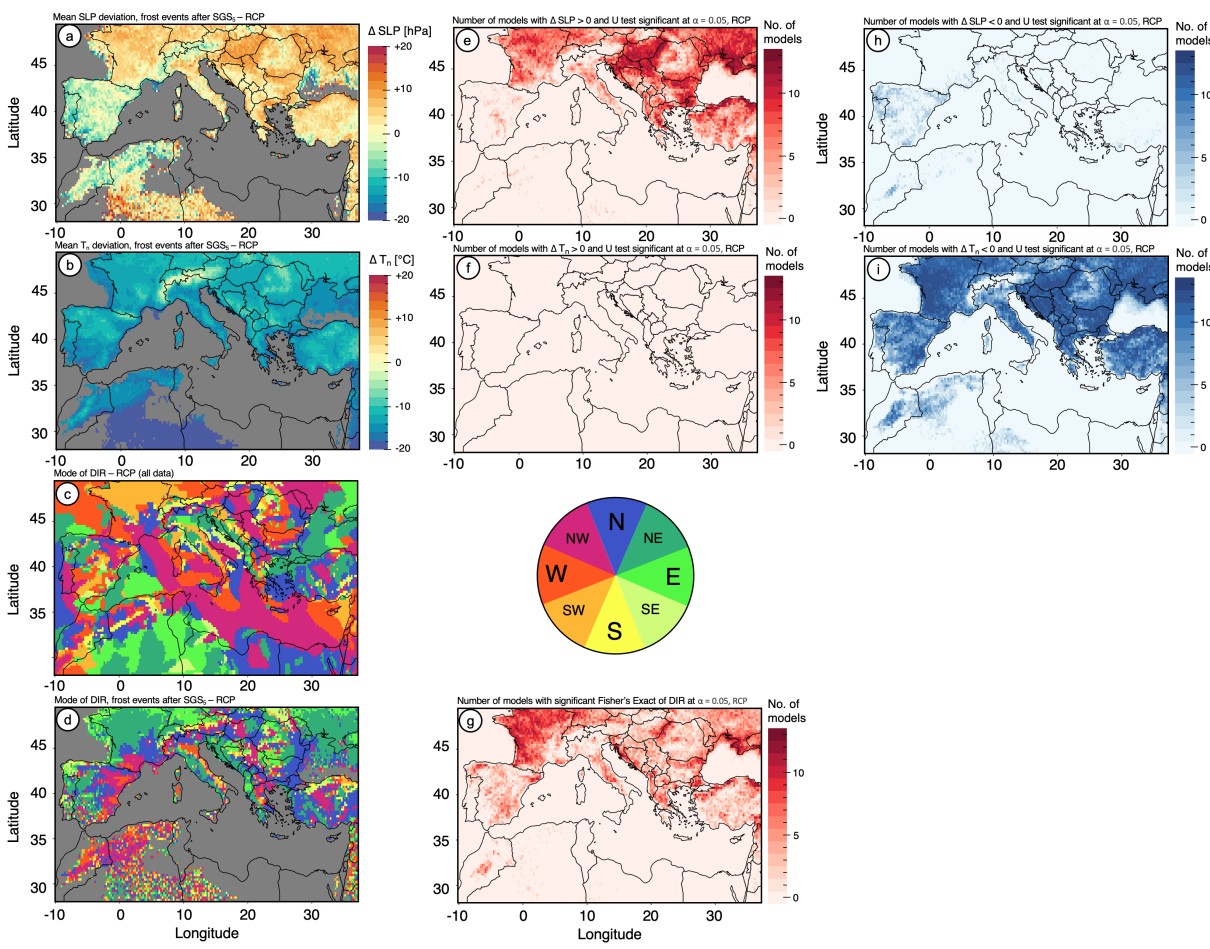

**Figure B2.** Same as B1, but for SGS$_5$.



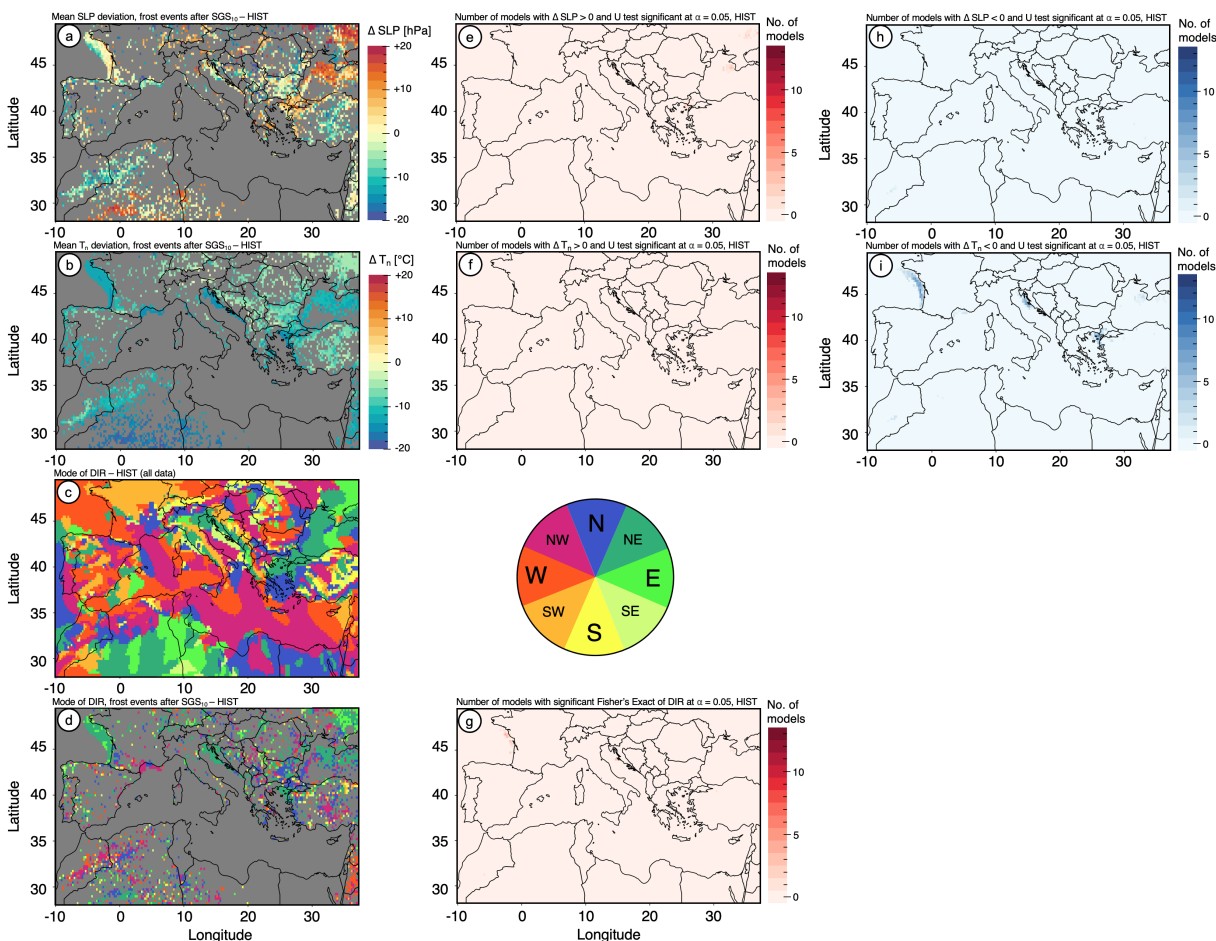

**Figure B3.** Same as B1, but for $SGS_{10}$ in the historical period.



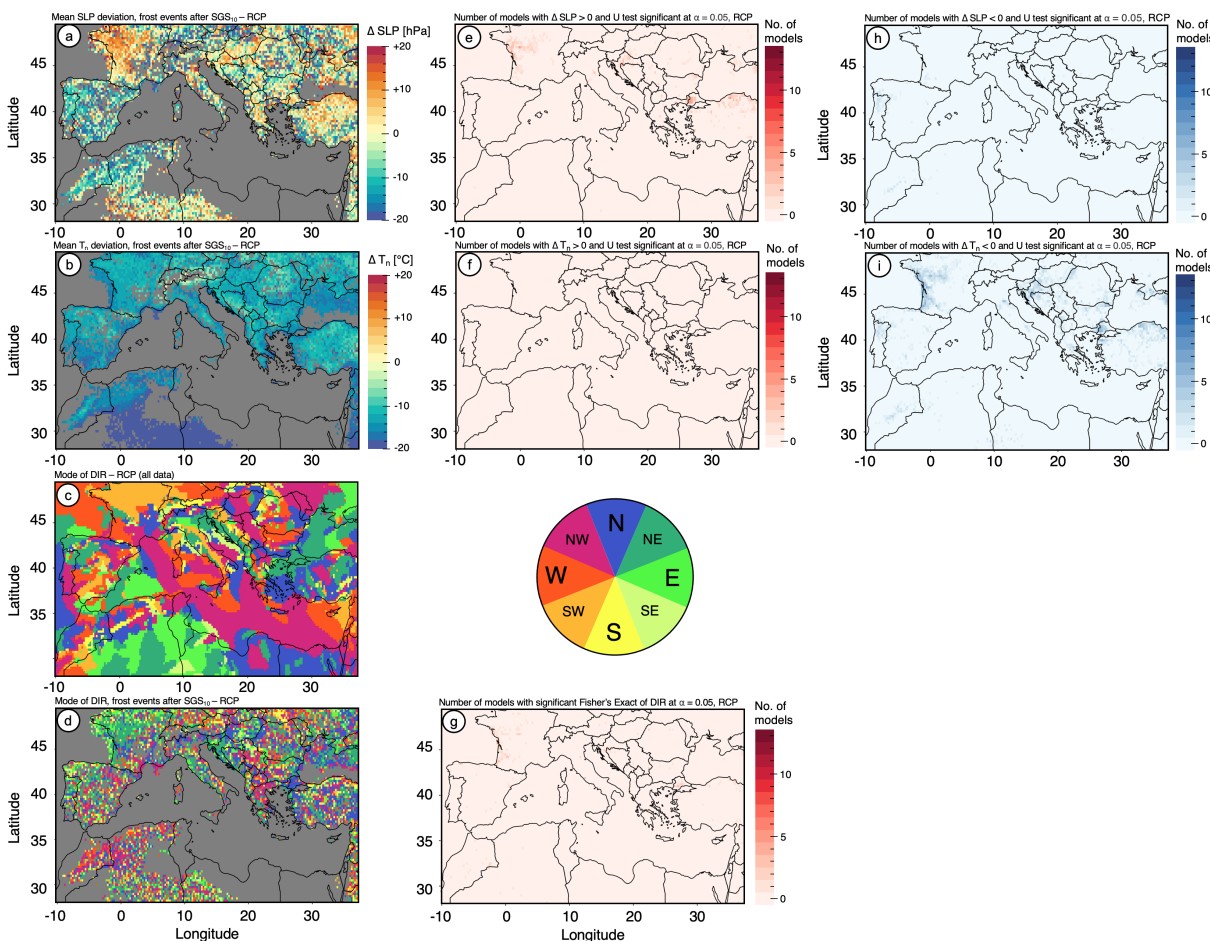

**Figure B4.** Same as B1, but for SGS$_{10}$ in the future period.





**Appendix C: Supplementary material for HDCEs**

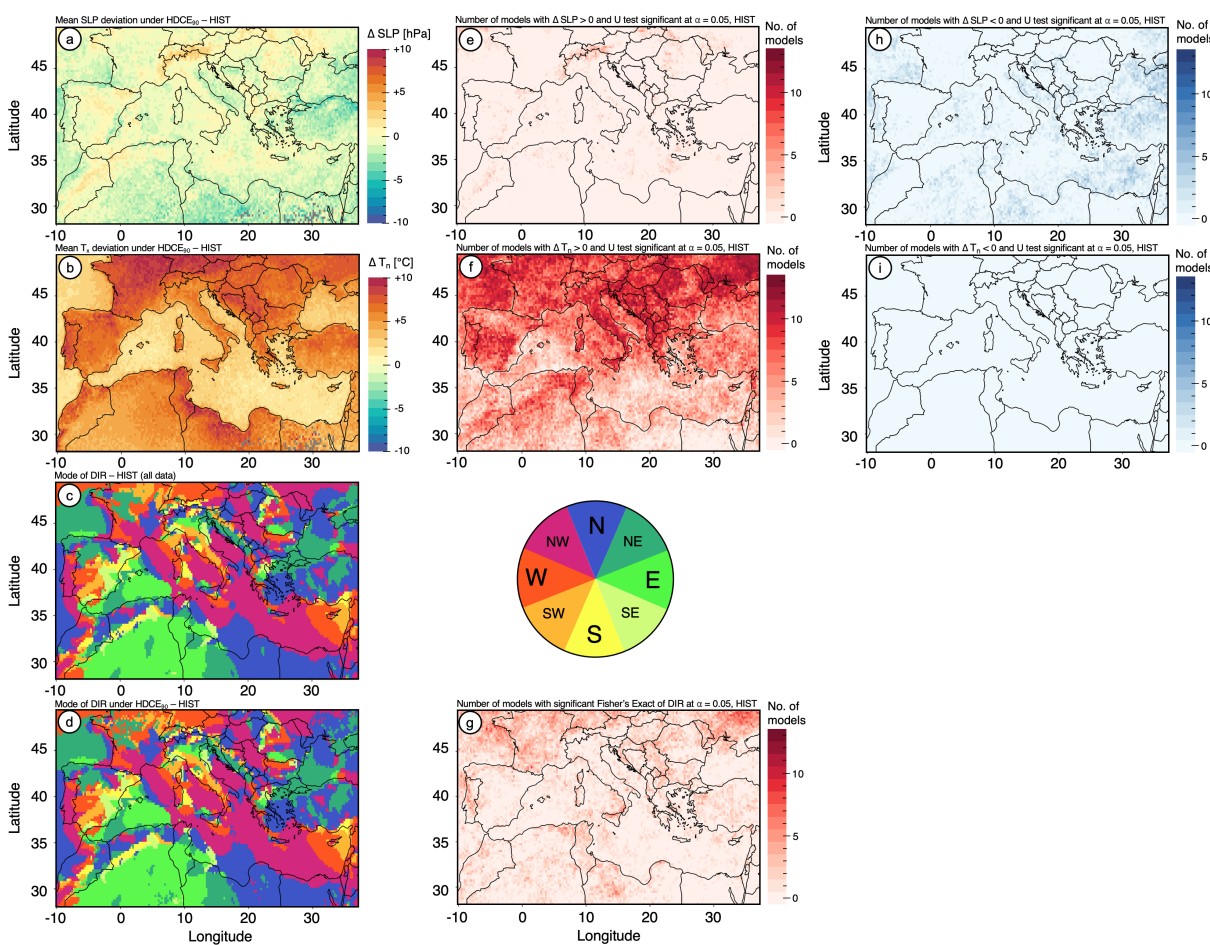

**Figure C1.** Mean deviation of the near-surface atmosphere from the 30-year mean under $HDCE_{90}$ for sea level pressure (SLP, first row) and daily maximum temperature ($T_x$, second row). Future deviations under RCP8.5. Number of models indicating a statistically significant positive deviation (e, f), and statistically significant negative deviation (h, i). Mode of wind direction within the 30-year future period (c) and only for HDCEs (d). Number of models indicating significantly differing distributions of wind directions (g).




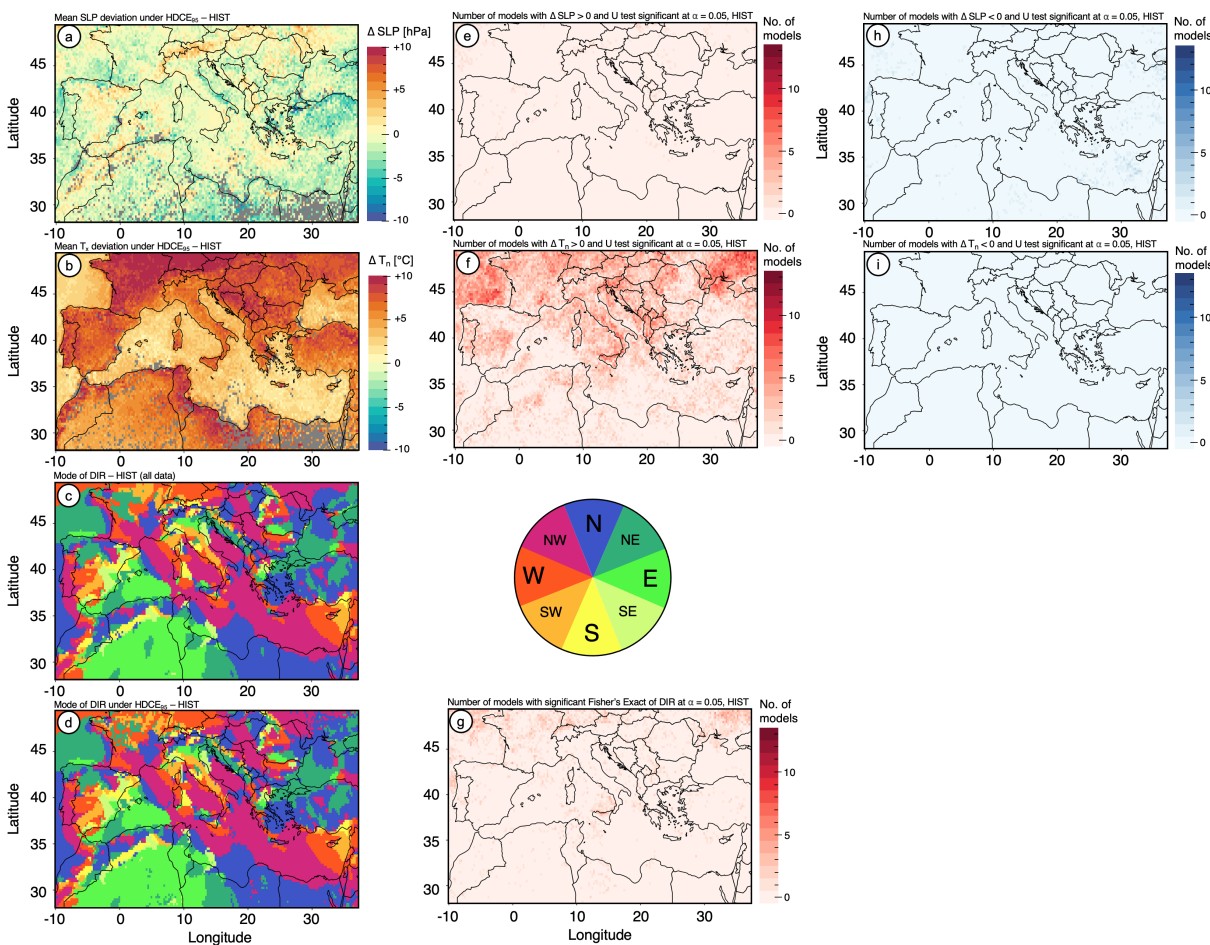

**Figure C2.** Same as C1, but for $HDCE_{95}$.


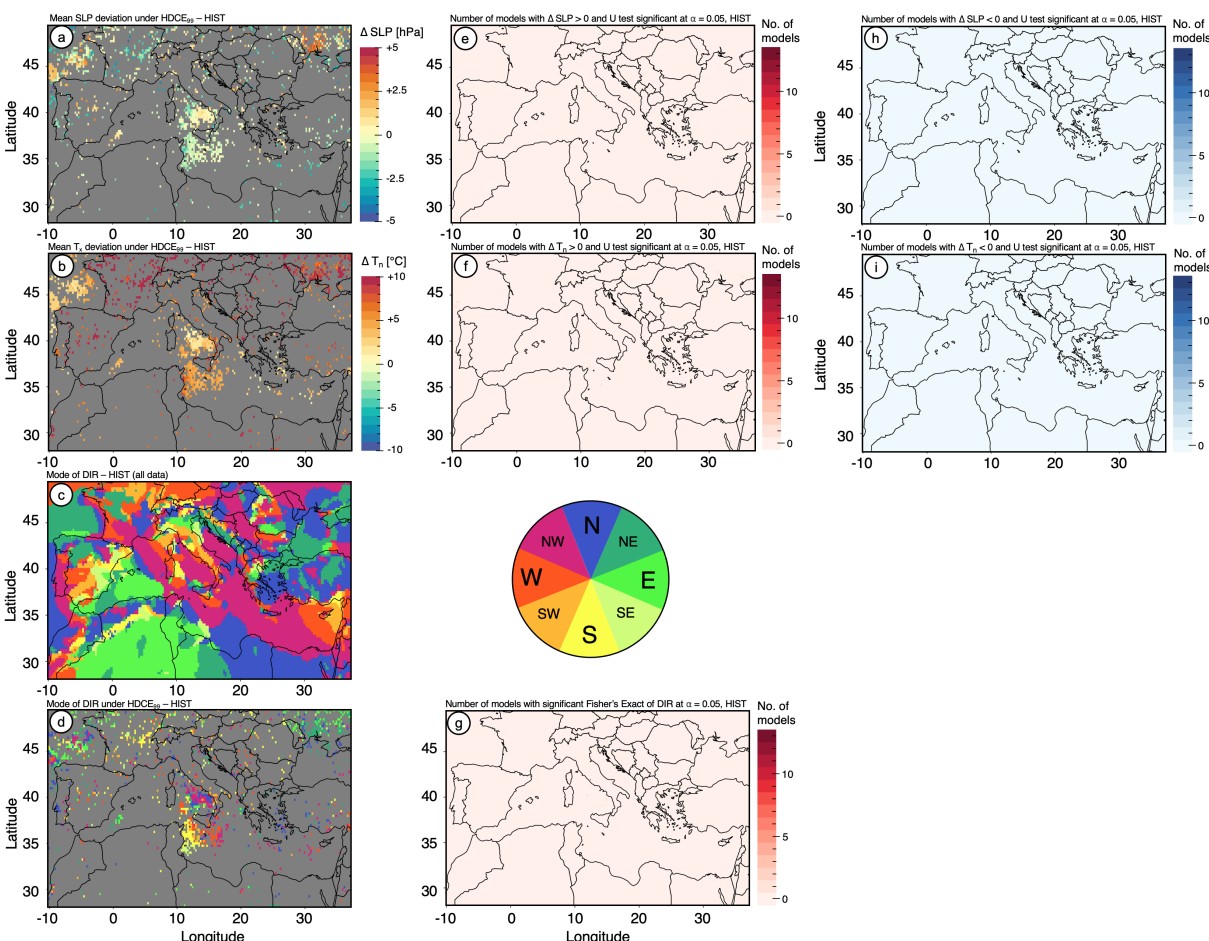

**Figure C3.** Same as C1, but for HDCE$_{99}$ in the historical period.





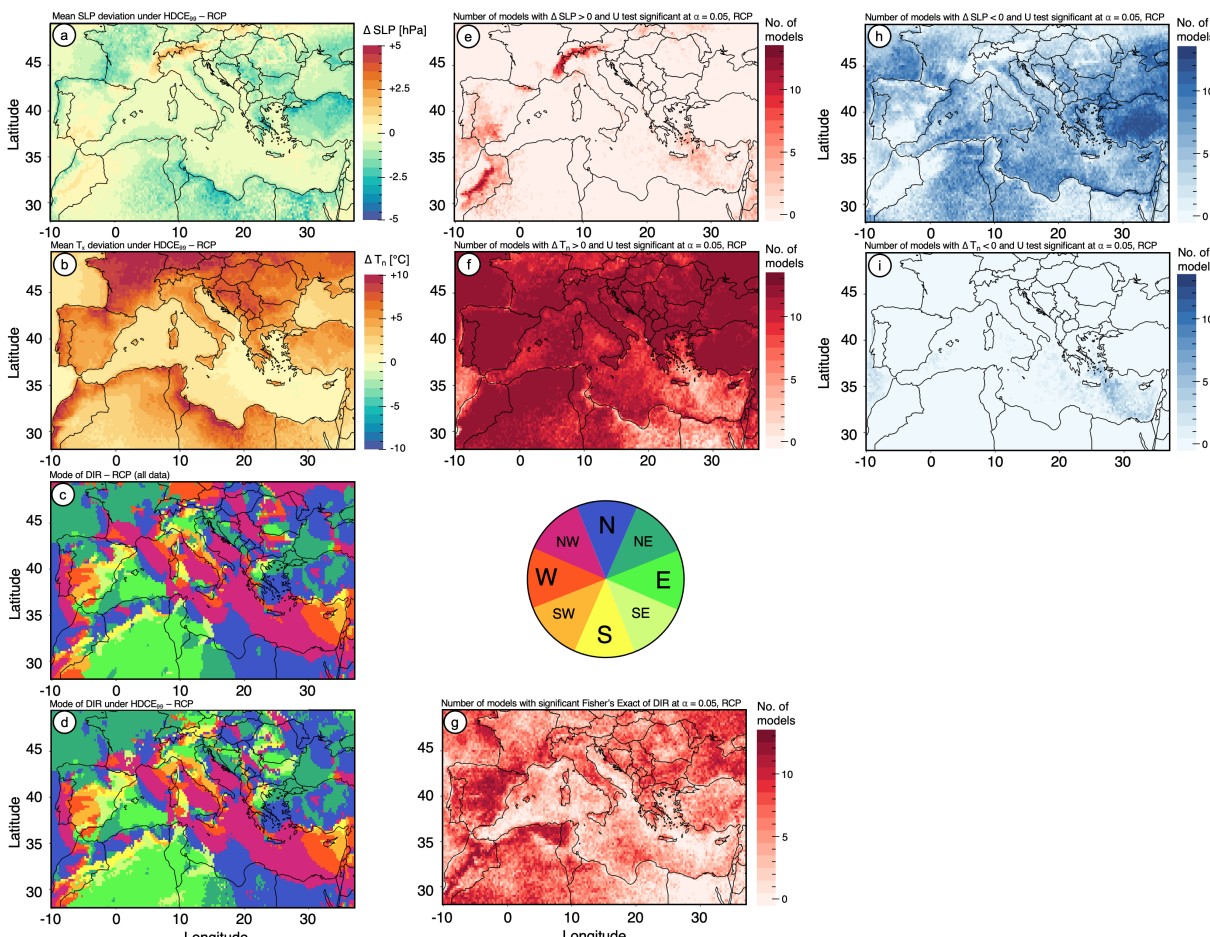

**Figure C4.** Same as C1, but for HDCE$_{99}$ in the future period.



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
