# Peer review of "Amplified potential for vegetation stress under climate change-induced intensifying compound extreme events in the Greater Mediterranean Region"

_Natural Hazards and Earth System Sciences, 2023_

## Author Comment (AC1)

Response to Anonymous Referee #1
Original comment in black, *response in blue and italics*

*We sincerely thank the two reviewers for their time and effort spent in helping us improve the quality of the manuscript. The main changes that have been made according to Anonymous Referee #1 's feedback include the following, and the specific comments are addressed below.*
- *We pointed out that the definition of the "Greater Mediterranean Domain" is purely based on the domain margins given within the CORDEX framework, on which this work is based on, and is not intended to resemble a climatic classification*
- *We extended explanations on the definitions of drought and heat*
- *We heavily shortened section 3 to exclude imprecise and redundant statements*

*Please note that all information on pages and lines refers to the track changes version of the revised manuscript.*

The paper by Olschewski et al. investigates the temporal changes of two types of compound effects (false spring events and heat-drought concurrence during summer) for the Mediterranean region comparing occurrences of the recent past (1970-1999) with a high-impact climate scenario for a time period hundred years later. Their study adds to the current knowledge of compound effects - which has previously done mostly done for past and current climatic setting – the changes of behaviour of a potentially extreme future. The study concludes with an important point which is often overlooked in climate adaption approaches: that locally refined investigations and adaptation strategies are required, as it is not correct to aggregate 'typical' climate patterns across larger parts of countries, as often the variations, in this case the intenseness of compound events, can be spatially very heterogeneous. In their study, they use data set from the EUR-CORDEX database and applied a multivariate bias correction method to improve spatial representation of future extremes. The analysis is very detailed regarding the definitions and thresholds being used for the two compound sets and spatially highly precise by presenting very detailed compound information on historical and projected changes.

The borders of the Greater Mediterranean Region are not clear. Does it depend on some climatic definition or simply latitude sub-sections? Northern France and Southern Germany would normally not be considered as part of the Mediterranean Region.

*The original idea for this study was to use the MED domain (Region 12, https://cordex.org/domains/region-12-mediterranean/ ) as provided within the CORDEX framework. However, the number of models provided for the EURO domain (Region 4) is significantly higher, granting more opportunities regarding the robustness of the results and the further use of the processed data sets. Therefore, we selected a domain similar to MED-CORDEX. This domain also includes large portions of central and eastern Europe that, from a climate classification perspective and as the reviewer suggests, are not considered as part of the Mediterranean Region. For the context of this study, we decided to use the term analogously to the definition within the CORDEX framework and added the term "Greater" to indicate that the investigations do not only address the climatological Mediterranean. We added two sentences in the Study domain section to make this demarcation clearer to the reader (l. 103-105).*

I would also like to ask the authors to discuss in more details the applicability of the HDCE definition: the results in Figure 9b for the future frequency of HCDE appear extremely large, however, it is after all mostly influenced by prolonged durations of future heatwaves under drought conditions. You may want to discuss your results in the light of other HDCE definitions.

*We agree that the changes in HDCE appear extremely large. The SPI-3 during summer alone is projected to decrease to -0.75 to -1 on average for large portions of the domain. This, combined with a projected increase in the temperature-based percentiles of 4.5 to 6 °C, causes a scenario with an extremely increased probability of an exceedance of the historical thresholds for HDCE.*

*We selected this specific definition of HDCE to keep the results comparable to the study by Ionita et al. (2021), who conducted similar investigations within a historical time period. The significance of the results would surely benefit from including other HDCE definitions, and we added a sentence in the discussion to point this out (l. 580-582).*

Overall, the paper is very well presented, both in terms of visual information, language and consistency of arguments. I recommend minor revision to include several smaller points listed below and summarised above.

*We would like to express our gratitude for the positive feedback.*

Minor comments

Second sentence in abstract: "it is highly dependent on the impact" seems to be semantically wrong, as the Medierraenan Basin cannot depend on impact. Please correct phrasing

*This is true. We changed the phrasing*

Line 116 to 122 appears very detailed, the last sentence of this paragraph appears out of place – I suggest to link the description of climate zone to an updated definition, what the GMR actually is (see comment above)

*Since the GMR, in our case, is not defined based on climatology, but the MED-CORDEX domain, we deemed it important to introduce the reader to all climatological areas within our domain limits.*

Line 124: not clear, what three types of cliamte data are meant: climate model output data, ERA5 data and ?

*Originally, the three types of climate data this sentence refers to were "ERA5", "historical climate model output", and "future climate model output". After reconsideration, we see that this sentence could be misleading. We have changed the sentence to indicate that 1. a reference data set and 2. climate model output must be obtained (l. 128-129).*

If ERA5 data are used to validate current runs of climate models, the weaknesses of ERA5 data (certain over- and underpredictions) should already be mentioned here, and not only in the discussion.

*We included two sentences within the introduction of the ERA5 data set to indicate weaknesses of ERA5 (l. 152-154).*

1 Data: you might want to mention that for your climate forcing analysis you actually need information from the ocean – it was first not clear why you cannnot use ERA5-Land data for evaluating agricultural developments.

*We have rephrased this sentence (l. 136).*

Line 146: this is an abrupt introduction of the 13 model runs – I would suggest to heavily shorten the section 124-136 and put all information on data in the 3.1 data section.

*We agree that this section can be heavily shortened. We moved all information on data into the 3.1 section and removed redundant phrasings.*

The same with Section 3.2: does this section need such a long, but imprecise introduction, and then additional sub-sections for compound event definitions. I would suggest to heavily shorten Lines 167-177)

*As above, we agree that this section can be shortened. We removed imprecise information on the bias correction method and the event definitions. We suggest to keep the information on the prevailing near-surface atmospheric conditions during these events within this section, since this analysis is independent from the specific compound event definitions presented in section 3.3.*

Section 3.2.2: how do you define an end of a drought? In McKees work, a drought ends, once the SPI value becomes from below -1 values again positive, i.e. includes values in the range of -1 to 0 if SPI had previously be below -1. This information is missed out in many papers – if you adopt another definition, you might want to state it more clearly.

*In fact, we did use a different definition for the end of a drought, i.e. the first day after a drought period for which the SPI-3 is higher than -1. We added a sentence to make this clearer (l. 288-289).*

Line 272: you might want to add here that there are my different definitions for heatwave events, and no clear definition available for heat compound events.

*We agree and have added a sentence to indicate the existing ambiguity in terms of heatwave definitions (l. 290-291).*

Lines 500 and 502: it is not the correct place in the article to discuss what is and what is not the aim of the study; you may want to consider to reword the two sentences.

*We removed the phrasing about the study aim, as the intention here is to point out a limitation of the study.*

You have chosen the most extreme climate projections – do you have any indications what the medium projections would say regarding the two compound events?

*We additionally performed the long-term climate assessment of the RCP 4.5 scenario in advance of preparing this manuscript. The bias corrected projections suggested an unfolding of climate change in the intermediate between no change signal and the RCP 8.5 scenario. The 30-year mean of maximum temperature is projected 1.6°C (RCP 4.5) / 3.7°C (RCP 8.5) warmer (2070-2099 minus 1990-2019), the annual precipitation sum is reduced by 2% (RCP 4.5) / 7% (RCP 8.5). At this point we decided to proceed using only RCP 8.5. Based on the long-term projections and a similar study conducted for a different domain (Olschewski et al. 2023), the projected changes in the inspected compound events may to similar proportions as for long-term climatology lie in the intermediate.*

Could you derive some hypthesis, what the combined effects of the increases of both compounds would have on agriculture or vegetation dynamics?

*Since the inspected compound events occur at different times throughout the year, crops may potentially be negatively affected multiple times within the crop life cycle. In regions for which an increase in both compound extremes is projected, this is disadvantageous for crops that are vulnerable to either one of the two, or in the worst case both. Crop damage occurring in spring may not be compensated until summer, and even if it is it may be lost when heat and drought occur. In this case, a stronger focus on more resilient crops may become necessary, next to crop protection in spring and effective irrigation in summer. However, making an assumption within the large-scale context of this study is difficult, especially in face of the plurality of involved factors, e.g., greatly varying resilience factors for individual crops, economic necessities and habits, technological advancement, ...*

*Literature*

*Ionita, Caldarescu, Nagavciuc (2021): Compound Hot and Dry Events in Europe: Variability and Large-Scale Drivers. Frontiers in Climate 3. https://doi.org/10.3389/fclim.2021.688991*

*Olschewski, Laux, Wei, Böker, Tian, Sun, Kunstmann (2023): An ensemble-based assessment of bias adjustment performance, changes in hydrometeorological predictors and compound extreme events in EAS-CORDEX. Weather and Climate Extremes 39. https://doi.org/10.1016/j.wace.2022.100531*

---

## Author Comment (AC2)

**Response to Anonymous Referee #2**
Original comment in black, *response in blue and italics*

*We sincerely thank the two reviewers for their time and effort spent in helping us improve the quality of the manuscript. The main changes that have been made according to Anonymous Referee #2 's feedback include the following, and the specific comments are addressed below.*

- *We included a list of all abbreviations in the appendix to better support the reader*
- *We extended the introduction of the MBCn method and added more information on the application of the method*

*Please note that all information on pages and lines refers to the track changes version of the revised manuscript.*

This study applied a MBCn to improve the model result by reducing the bias compared to Univariate. The results showed how the FSE and HDCEs trend changed under the high impact scenario. Overall, the paper is clear and well presented. However, there is still some minor issues need to be addressed. *Thank you for the positive feedback.*

Line 270, I recommend heatwave events abbreviation could be used as HWE or HE, not HW. *We changed the abbreviation to HWE throughout the manuscript.*

Since the manuscript has so many abbreviations, it is better to create table or separate page as appendix to list them together to be easier to follow. Also, please define each acronym at the very beginning. *We included a list of all abbreviations in the appendix. The calculations of the abbreviated variables are presented in Table 2. Additionally, we revised the manuscript to ensure all abbreviations are explained in the text when they first appear.*

The MBCn method section needs more details, since it is important for readers to understand how method is applied. *We extended section 3.2.2 Multivariate bias correction to include more information on the application of MBCn and additional possibilities within this framework presented by Cannon 2016.*

*Literature*
*Cannon (2016): Multivariate Bias Correction of Climate Model Output: Matching Marginal Distributions and Intervariable Dependence Structure. Journal of Climate 29 (19). https://doi.org/10.1175/JCLI-D-15-0679.1*